# Model Inversion with Layer-Specific Modeling and Alignment for Data-Free Continual Learning

**Ruilin Tong[1], Haodong Lu[1], Yuhang Liu[2], Dong Gong[1]**[*]

[1] School of Computer Science and Engineering, University of New South Wales
[2] Australian Institute for Machine Learning, The University of Adelaide
`{ruilin.tong, haodong.lu, dong.gong}@unsw.edu.au`
`yuhang.liu01@adelaide.edu.au`

## Abstract

Continual learning (CL) aims to incrementally train a model to a sequence of tasks while maintaining performance on previously seen ones. Despite mitigating forgetting, data storage and replay are often infeasible due to privacy or security constraints and are impractical for arbitrary pre-trained models. Data-free or examplar-free CL aims to continually update models with new tasks without storing previous data. In addition to regularizing updates, we employ model inversion to synthesize data from the trained model, anchoring learned knowledge through replay without retaining old data. However, model inversion in predictive models faces two key challenges. First, generating inputs (e.g., images) solely from highly compressed output labels (e.g., classes) often causes drift between synthetic and real data. Replaying on such synthetic data can contaminate and erode knowledge learned from real data, further degrading inversion quality over time. Second, performing inversion is usually computationally expensive, as each iteration requires backpropagation through the entire model and many steps are needed for convergence. These problems are more severe with large pre-trained models such as Contrastive Language-Image Pre-training (CLIP) models. To improve model inversion efficiency, we propose Per-layer Model Inversion (PMI) approach inspired by the faster convergence of single-layer optimization. The inputs optimized from PMI provide strong initialization for full-model inversion, significantly reducing the number of iterations required for convergence. To address feature distribution shift, we model class-wise feature distribution using a Gaussian distribution and preserve distributional information with a contrastive model. Sampling features for inversion ensures alignment between synthetic and real feature distributions. Combining PMI and feature modeling, we demonstrate the feasibility of incrementally training models on new classes by generating data from pseudo image features mapped through semantic-aware feature projection. Our method shows strong effectiveness and compatibility across multiple CL settings. Code is available at `https://github.com/RuilinTong/PMI-CFS-DFCL`.

## 1   Introduction

In real-world scenarios, intelligent agents are required to learn from evolving data presented as a sequence of tasks. However, models often forget previously acquired knowledge when trained on new tasks — a phenomenon known as catastrophic forgetting [28]. Continual learning (CL) aims to address this challenge by enabling models to incrementally learn new tasks while preserving knowledge from earlier tasks [36]. A common strategy to mitigate this is to store a subset of data

---

[*]D. Gong is the corresponding author.

39th Conference on Neural Information Processing Systems (NeurIPS 2025).

from previous tasks and replay it as memory during training to preserve learned knowledge [35, 7, 43]. However, storing previous data may be infeasible—particularly due to privacy or security constraints. Data-free (or exemplar-free) CL [54, 29, 40, 38] aims to alleviate catastrophic forgetting without storing previous data. Apart from regularizing updating [54, 22, 24, 29], model inversion is an alternative that generates synthetic data by extracting knowledge from the trained model to anchor the learned knowledge through replay [60, 39, 11], while learning new tasks.

Applying model inversion to CL presents two key challenges. First, generating images solely from integer class labels and classification loss [39, 11] can lead to a distributional mismatch. Specifically, there exist multiple feature representations that can yield low classification loss. As a result, the features of synthetic data may deviate from those of real images, indicating the generated samples contain information that differs from the real data. This sample information drift introduces incorrect or mismatched knowledge, which can harm CL — the model may forget real image knowledge after replaying low-quality (misaligned) synthetic data. Moreover, because model inversion relies on knowledge encoded in the model itself, such forgetting further degrades the quality of future inversions in subsequent tasks. CLIP models can substantially enhance CL performance due to their strong zero-shot generalizability [16, 61, 27]. However, replaying synthetic data whose distribution differs from real data can also weaken the pre-trained knowledge of CLIP models. Second, model inversion is computationally expensive. The process involves iteratively updating input images so that the model recognizes them as belonging to specific classes. Each update step requires backpropagating gradients through the entire model, and many iteration steps are needed for convergence. Prior works [60, 41, 13] primarily focus on CNN-based architectures, where the computational cost is more manageable. Hu et al. [14] reduce this cost by selectively dropping patches during inversion. However, a large number of iteration steps are still required for convergence. These problems become more severe when applying model inversion to large pre-trained foundation models such as CLIP models. Beyond generating replay data for previous tasks, model inversion guided by text prompts can also synthesize images from CLIP by leveraging its rich pre-trained knowledge [19]. In CLIP-based CL, this enables adaptation to new classes without collecting additional training data, by generating synthetic samples based on corresponding text features.

To address the computational cost of model inversion, we propose Per-layer Model Inversion (PMI), which reconstructs optimal layer inputs in a top-down, layer-by-layer manner. The key insight is that the loss landscape of an individual layer is significantly simpler than that of the full model, allowing the intermediate representations to converge more efficiently. The inputs obtained through PMI serve as good initializations for full-model inversion, thereby reducing the number of update steps required for convergence. To mitigate feature distribution shift, we propose to model the feature distribution of each class and sample features as the targets for inversion such that the features of synthetic data remain aligned with the distribution of real features. While a Gaussian distribution is commonly used to approximate class-specific feature distributions, we further introduce a lightweight contrastive model trained on real data features using a negative contrastive loss. This model better preserves the underlying structure of the feature space. During inversion, features are sampled from the Gaussian distribution and filtered by the contrastive model, leveraging the information it encodes to guide the selection toward more realistic representations. Building on PMI and feature distribution modeling, we propose to incrementally train the model on new classes without collecting real training data by generating pseudo image features through semantic-aware feature projection, from which training samples are synthesized via model inversion.

We summarize our contribution as follows:

- We address model inversion inefficiency by introducing PMI, which significantly reduces the number of optimization steps. Our empirical results demonstrate both the efficiency of PMI and its compatibility with various baseline methods.

- We address feature distribution shift between synthetic and real data by modeling class-wise feature distributions using Gaussian and lightweight contrastive models, from which features are sampled for inversion. Our approach consistently improves performance across different models e.g., ResNet and CLIP model.

- We demonstrate the feasibility of training models on new classes without collecting additional training data, by generating synthetic samples of new classes from the CLIP model using model inversion and semantic-aware feature projection. Experimental results validate the effectiveness of our approach.

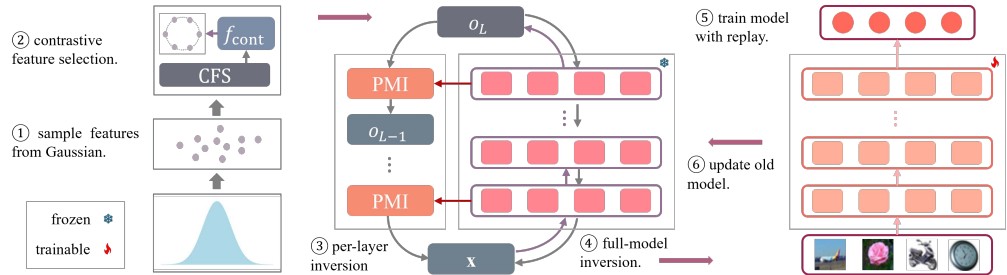

Figure 1: Overview of our model inversion method for data-free CL. Class-wise features are sampled from Gaussian distributions and selected using the CFS strategy (Steps ① and ②, Section 3.3). Based on the selected features, synthetic data is generated via our PMI + full-model inversion method (Steps ③ and ④, Section 3.2). In Step ⑤, the model is trained on the new task using replay, and the inversion model is updated after each task (Step ⑥).

## 2 Related work

**Model inversion.** Model inversion aims to recover training data from trained models and has been widely applied in data-free knowledge transfer [63, 52, 44, 41], data-free CL [39, 11, 33, 1], and meta-learning [13, 15, 56]. [19] use model inversion to analyze knowledge encoded in CLIP models, while [55] apply it for open-vocabulary customization. Although effective, model inversion is computationally expensive. Sparse-MI [14] reduces this cost by progressively dropping image patches in ViT models, but still requires a large number of iterations to convergence.

**Continual learning.** Continual learning aims to incrementally train models to a sequence of tasks while retaining performance on previously learned tasks, typically without access to prior task data. Existing approaches include regularization-based methods [22, 64, 6, 37, 24, 30, 58], parameter-isolation methods [59, 47, 66, 18, 31, 3, 48], and rehearsal-based methods [26, 35, 4, 5, 58, 43]. While rehearsal methods are effective, storing past data is often infeasible due to privacy or security constraints. To address this, prior works [60, 39, 11, 33, 1] use model inversion to generate synthetic data for data-free CL. However, label-based inversion guided by classification loss can lead to distribution shift between synthetic and real features, degrading CL performance.

**Continual learning on pre-trained models.** Pre-trained vision transformers (ViTs) exhibit strong generalization to unseen domains [62], and parameter-efficient methods [29, 40, 53, 54, 25, 48] have successfully adapted ViTs for CL, significantly boosting performance. Vision-language models (VLMs) such as CLIP [34] and Flamingo [2] demonstrate strong zero-shot capabilities and have been applied in CL settings [42, 9]. To efficiently adapt CLIP for CL, recent methods adopt prompt-based [20, 68, 50] or adapter-based [10, 65, 27] tuning strategies. Beyond tuning, the rich knowledge in CLIP models can also be leveraged to synthesize training images via model inversion.

## 3 Model inversion for data-free CL

### 3.1 Preliminaries

**Model inversion for data-free CL.** Data-free (or exemplar-free) CL [38, 40, 29] aims to continually train a model while preserving performance on previously learned tasks without storing previous data. During training on task $t$, the model does not have access to data from previous tasks. We denote the dataset and class set of task $t$ as $\mathcal{D}_t$ and $\mathcal{C}_t$, respectively, with no class overlap between different tasks. Model inversion generates previous data for replay directly from the trained model, this is done by fixing the model parameters and optimizing the input $\mathbf{x}$, so that the model predicts a given target class $y$. In this work, we consider a standard CL model composed of a backbone with totally $L$ layers and a classification head. We refer to the output of the backbone as feature in our work.

**CLIP-based CL.** CLIP models consist of an image encoder $F_i$ and a text encoder $F_t$ trained jointly on millions of image-text pairs using a contrastive loss. For classification tasks, $F_i$ serves as backbone model, while the text features extracted by $F_t$ function as the classification head. $\mathbf{t}_c$ denotes the input text for class $c$, composed of hand-crafted prompts combined with the class name.

**Overview.** We address the high computational cost of model inversion and propose PMI in Section 3.2, which significantly reduces the number of update steps. In Section 3.3, we mitigate feature distribution shift by modeling real feature distributions using both Gaussian distribution and contrastive models. Features are sampled from these distributions as the targets for inversion, ensuring that the features of synthetic data align with the distribution of real features. Finally, in Section 3.4, we explore the potential of generating training samples for new classes from pseudo image features generated by semantic-aware feature projection, leveraging the pre-trained knowledge of the CLIP model. Figure 1 presents an overview of our model inversion method for data-free CL. Detailed algorithms and loss functions are provided in Appendix C.

## 3.2 Layer-wise alignment for efficient model inversion

In this section, we first formulate the model inversion objective using KL divergence and decompose it into layer-wise constraints. Based on this formulation, we propose a layer-by-layer inversion approach. We show that this decomposed objective is equivalent to prior model inversion objectives without smoothness constraints in full-model inversion case. The key insight is that the loss landscape of a single layer is significantly simpler than that of the full model, enabling faster convergence. The synthetic input optimized by PMI serves as a strong initialization for full-model inversion, reducing the number of iterations required for convergence.

VMI [49] formulates the model inversion objective using KL divergence, aiming to recover the sampling distribution of real data. DeepInversion [60] proposed to further constrain the distribution of intermediate layer outputs of synthetic data to match a prior distribution. Let $\mathbf{o}_l$ denote the output of the $l$-th layer, $\mathbf{x}$ the input and $y$ the label and $L$ the total number of layers in the backbone model. Our goal is to recover the joint distribution over inputs and intermediate outputs, which we formulate as a KL divergence objective:

$$p_s^*(\mathbf{x}) = \arg\min_{p_s(\mathbf{x})} D_{\mathrm{KL}}(p_s(\mathbf{o}_L, \mathbf{o}_{L-1}, \cdots, \mathbf{x}|y) || p_r(\mathbf{o}_L, \mathbf{o}_{L-1}, \cdots, \mathbf{x}|y)), \tag{1}$$

where $p_r(\cdot)$ denotes the probabilities computed on real data and $p_s(\cdot)$ denotes the probabilities computed on synthetic data. Assuming that the synthetic $\mathbf{o}_l$ contains sufficient information to reconstruct $\mathbf{o}_{l-1}, \cdots, \mathbf{x}$ and that the influence of $\mathbf{o}_{l+1}, \cdots, y$ can be neglected during this reconstruction, and noting that $\mathbf{o}_0 = \mathbf{x}$, the objective in Eq. (1) can be decomposed into layer-wise constraints as:

$$D_{\mathrm{KL}}(p_s(\mathbf{o}_L, \mathbf{o}_{L-1}, \cdots, \mathbf{x}|y) || p_r(\mathbf{o}_L, \mathbf{o}_{L-1}, \cdots, \mathbf{x}|y))$$
$$\approx \sum_{l=1}^{L} \mathbb{E}_{p_s(\mathbf{o}_{l+1}|y)} \left[ \mathbb{E}_{p_s(\mathbf{o}_l|\mathbf{o}_{l+1})} D_{\mathrm{KL}}(p_s(\mathbf{o}_{l-1}|\mathbf{o}_l) || p_r(\mathbf{o}_{l-1}|\mathbf{o}_l)) \right] + D_{\mathrm{KL}}(p_s(\mathbf{o}_L|y) || p_r(\mathbf{o}_L|y)). \tag{2}$$

Namely, minimizing objective in Eq. (1) is equivalent to minimizing input constraint at each layer. The detailed derivation is provided in Appendix A.

However, the probabilities in the decomposed KL divergence are intractable in practice. To obtain a tractable objective, we follow [49] and apply Bayes' rule to transform each KL term as:

$$D_{\mathrm{KL}}(p_s(\mathbf{o}_{l-1}|\mathbf{o}_l) || p_r(\mathbf{o}_{l-1}|\mathbf{o}_l))$$
$$= D_{\mathrm{KL}}(p_s(\mathbf{o}_{l-1}|\mathbf{o}_l) || p_r(\mathbf{o}_{l-1})) - \mathbb{E}_{p_s(\mathbf{o}_{l-1}|\mathbf{o}_l)} \log p_r(\mathbf{o}_l|\mathbf{o}_{l-1}) + \log p_r(\mathbf{o}_l), \tag{3}$$

The negative log-probability term $-\log p_r(\mathbf{o}_l|\mathbf{o}_{l-1})$ is approximated using mean squared error (MSE). For final term $D_{\mathrm{KL}}(p_s(\mathbf{o}_L|y) || p_r(\mathbf{o}_L|y))$, the negative log-probability is approximated by cross-entropy (CE) loss for classification tasks, and by MSE for regression tasks. For the prior constraint $D_{\mathrm{KL}}(p_s(\mathbf{o}_{l-1}|\mathbf{o}_l) || p_r(\mathbf{o}_{l-1}))$, we follow ABD [39] and approximate both distributions as Gaussians, allowing closed-form computation of the KL divergence. If $\mathbf{o}_l$ is known, the expectation over $p_s(\mathbf{o}_{l+1}|y) p_s(\mathbf{o}_l|\mathbf{o}_{l+1})$ can be approximated by average. To compute $\mathbf{o}_l$, we propose Per-layer Model Inversion (PMI), which minimizes the layer-wise constraint from the output layer to the input layer-by-layer, based on the formulation in Eq. (2) and the approximations discussed above. After optimizing the input to the $(l+1)$-th layer to convergence, the resulting synthetic input is used for inverting the $l$-th layer. The resulting inversion loss for each layer is:

$$\mathbb{E}_{p_s(\mathbf{o}_{l+1}|y)} \left[ \mathbb{E}_{p_s(\mathbf{o}_l|\mathbf{o}_{l+1})} D_{\mathrm{KL}}(p_s(\mathbf{o}_{l-1}|\mathbf{o}_l) || p_r(\mathbf{o}_{l-1}|\mathbf{o}_l)) \right]$$
$$\approx D_{\mathrm{KL}}(\mathcal{N}(\hat{\boldsymbol{\mu}}_{l-1}, \hat{\boldsymbol{\sigma}}_{l-1}) || \mathcal{N}(\boldsymbol{\mu}_{l-1}, \boldsymbol{\sigma}_{l-1})) + \frac{1}{N} \sum_{i=1}^{N} \ell_{\mathrm{MSE}}(\mathbf{o}_{l-1,i}, \mathbf{o}_{l,i}; \boldsymbol{\theta}_l), \tag{4}$$

where $\boldsymbol{\mu}_{l-1}$ and $\boldsymbol{\sigma}_{l-1}$ denote the mean and standard deviation of $\mathbf{o}_{l-1}$ computed from real data, while $\hat{\boldsymbol{\mu}}_{l-1}$ and $\hat{\boldsymbol{\sigma}}_{l-1}$ are computed from synthetic features, and $\boldsymbol{\theta}_l$ represents the parameters of the $l$-th layer. Eq. (4) shows that optimizing the layer input aims to keep it within the prior distribution while aligning the corresponding layer output to a given target. The overall inversion objective across all layers is given by:

$$D_{\mathrm{KL}}(p_{\mathrm{s}}(\mathbf{o}_L, \mathbf{o}_{L-1}, \cdots, \mathbf{x}|y)||p_{\mathrm{r}}(\mathbf{o}_L, \mathbf{o}_{L-1}, \cdots, \mathbf{x}|y))$$
$$\approx \sum_{l=1}^{L} \left( D_{\mathrm{KL}}(\mathcal{N}(\hat{\boldsymbol{\mu}}_{l-1}, \hat{\boldsymbol{\sigma}}_{l-1})||\mathcal{N}(\boldsymbol{\mu}_{l-1}, \boldsymbol{\sigma}_{l-1})) + \frac{1}{N} \sum_{i=1}^{N} \ell_{\mathrm{MSE}}(\mathbf{o}_{l-1,i}, \mathbf{o}_{l,i}; \boldsymbol{\theta}_l) \right) \quad (5)$$
$$+ D_{\mathrm{KL}}(\mathcal{N}(\hat{\boldsymbol{\mu}}_L, \hat{\boldsymbol{\sigma}}_L)||\mathcal{N}(\boldsymbol{\mu}_L, \boldsymbol{\sigma}_L)) + \frac{1}{N} \sum_{i=1}^{N} \ell_{\mathrm{CE}}(\mathbf{o}_{L,i}, y; \boldsymbol{\theta}_{L+1}).$$

For the detailed derivation, please refer to Appendix B.

Performing model inversion on the full model neglects the loss terms $\ell_{\mathrm{MSE}}(\mathbf{o}_{l-1,i}, \mathbf{o}_{l,i}; \boldsymbol{\theta}_l)$ as there is no target $\mathbf{o}_{l,i}$ available for the output of the $(l-1)$-th layer, and each $\mathbf{o}_l$ is computed from $\mathbf{x}$. Eq (5) is equivalent to the full-model inversion objective proposed in ABD [39], excluding the smoothness constraint.

The key insight of PMI is that the loss landscape of a single layer is significantly simpler than that of the full model, as shown in Appendix G, which allows convergence in fewer update steps and yields low final loss during single-layer inversion. Since each layer-wise inversion converges with minimal error in Eq. (4),

---

**Algorithm 1:** PMI+full-model inversion

**Input:** Trained model with parameter $\boldsymbol{\theta}$,
Class feature for inversion $\mathbf{o}_L$.
**Output:** Synthetic input $\hat{\mathbf{x}}$
**for** $l = L$ **to** $1$ **do**
   Randomly initialize input of $l$-th layer
    $\hat{\mathbf{o}}_{l-1}$;
   Update $\hat{\mathbf{o}}_{l-1}$ with objective in Eq. (4);
   Set $\hat{\mathbf{o}}_{l-1}$ as target output for layer $l-1$;
$\hat{\mathbf{x}} = \hat{\mathbf{o}}_0$;
Finetune $\hat{\mathbf{x}}$ with full-model inversion objective
  in Eq. (5);

---

the accumulated error across layers in Eq. (5) remains small, making the inputs optimized by PMI close to optimal. To further correct accumulated approximation errors, we apply few fine-tuning steps on the full model after completing PMI. As the inputs from PMI serve as a strong initialization, fewer iterations are required for convergence during full-model inversion. We refer to this overall strategy as PMI+full-model inversion.

For ResNets, each residual block is treated as a single layer, while for ViTs used in CLIP models, each transformer block is considered one layer. Unlike ResNets, ViTs lack Batch Normalization layers and thus do not store input statistics. To enable layer-wise distribution constraints in CLIP models, we compute and store input statistics from real data for use in Eq. (4) and Eq. (5). Specifically, the stored statistics consist of the mean and standard deviation of layer inputs. These values are highly compressed and contain no identifiable information about the original data, ensuring that the method remains both practical and fully compliant with the data-free setting.

### 3.3 Class-wise feature modeling and sampling for real-synthetic feature alignment

In classification models, mapping a high-dimensional vector $\mathbf{o}_L$ to an integer label $y$ discards a substantial amount of information encoded in $\mathbf{o}_L$. As a result, the rich semantic content in $\mathbf{o}_L$ cannot be reliably recovered from a single label and the CE loss alone. Specifically, multiple $\mathbf{o}_L$ values drawn from different distributions may yield low CE loss for the same label $y$. Our experiments shown in Appendix E demonstrate a distribution shift between synthetic and real feature representations, suggesting that the synthetic data may encode information that differs from that of real data. In data-free CL, replaying synthetic data serves to remind the model of knowledge from previous tasks. However, if the synthetic data encodes information that deviates from the real data, it may

---

**Algorithm 2:** Contrastive feature selection.

**Input:** Gaussian distribution $\mathcal{N}(\boldsymbol{\mu}_c, \boldsymbol{\sigma}_c)$ of
     class $c$, contrastive model $f_{\mathrm{cont}}(\phi)$,
     selection steps $n$
**Output:** Feature set for inversion $\mathcal{S}_{\mathrm{feat}}$
Initialize $\mathcal{S}_{\mathrm{feat}}$ from $\mathcal{N}(\boldsymbol{\mu}_c, \boldsymbol{\sigma}_c)$;
**for** $i = 1$ **to** $n$ **do**
   Sample $m$ features from $\mathcal{N}(\boldsymbol{\mu}_c, \boldsymbol{\sigma}_c)$;
   Compute $\mathcal{L}_{\mathrm{cont}}(\mathbf{o}_{L,j}, \mathcal{S}_{\mathrm{feat}}; f_{\mathrm{cont}})$ for each
    $\mathbf{o}_{L,j}, j \in [1, m]$;
   Select features with lowest
    $\mathcal{L}_{\mathrm{cont}}(\mathbf{o}_{L,j}, \mathcal{S}_{\mathrm{feat}}; f_{\mathrm{cont}})$ and add into $\mathcal{S}_{\mathrm{feat}}$;

---

cause the model to forget previously learned knowledge after replay. Moreover, since model inversion relies on extracting knowledge from the model itself, this forgetting further degrades the quality of model inversion in future stages. Since CLIP models encode rich semantics in the feature space, feature difference between real and synthetic data indicates a substantial semantic change in the synthetic data. Training on these misaligned samples—where labels no longer align with the underlying semantics—can further degrade the pre-trained knowledge of CLIP models as discussed in Appendix D.4.

To address this issue, we propose modeling the class-wise real feature distribution and sampling features from it as the targets for inversion, ensuring that synthetic data features lie within the distribution of real features. Specifically, PMI is initiated from the sampled feature representation $\mathbf{o}_L$ rather than the class label $y$, namely the CE loss term in Eq. (5) is removed, and the inversion process aims to align the features of the generated input with the sampled $\mathbf{o}_L$. A simple yet effective approach is to model the class-wise real feature distribution using a Gaussian distribution. However, a Gaussian distribution captures only the mean and variance of real features, potentially neglecting other important aspects of the feature distribution. To better preserve this distributional information, we train a lightweight MLP, referred to as the contrastive model, on class-wise real features using a negative contrastive loss

$$\mathcal{L}_{\text{cont}}(\mathbf{o}_{L,i}, \mathcal{S}_{\text{neg}}; f_{\text{cont}}) = \log \ \mathbb{E}_{\mathbf{o}_{L,j} \in \mathcal{S}_{\text{neg}}} \left[ e^{\cos(f_{\text{cont}}(\mathbf{o}_{L,i}), f_{\text{cont}}(\mathbf{o}_{L,j}))} \right], \quad \mathbf{o}_{L,j} \neq \mathbf{o}_{L,i}, \tag{6}$$

where $\cos(\cdot, \cdot)$ denotes cosine similarity, $\mathcal{S}_{\text{neg}}$ is a subset of real features excluding the $i$-th feature $\mathbf{o}_{L,i}$, and serves as a negative set in the contrastive loss. $f_{\text{cont}}$ represents the contrastive model. Optimizing Eq. (6) aims to map all real features onto a hyper-sphere uniformly as discussed in [51]. Uniform mapping ensures that no feature is overly compressed, thereby preserving the maximum amount of information from the original feature distribution. Previous works [51, 46] introduce a temperature parameter in the contrastive loss to control the sharpness of the similarity distribution. In our work, we omit this term due to its minimal effect, as discussed in detail in Appendix D.3.

To leverage the information encoded in the contrastive model, we aim to select a set of features that can be mapped uniformly to a hypersphere by contrastive model, indicating maximally recovered distributional information. To achieve this, we adopt a greedy incremental selection strategy, referred to as Contrastive Feature Selection (CFS). Starting from an initial random set, we sample candidate features from the Gaussian distribution and map both the selected and candidate features using the contrastive model. At each step, we select the candidate feature that has the minimum average cosine similarity with the currently selected features in the mapped space. Through multi-step selection, the selected features can be mapped uniformly on the hypersphere, indicating that the selected features capture the maximum information from the real feature distribution. Algorithms for refined model inversion strategy and CFS are shown in Alg. 1 and Alg. 2 respectively.

### 3.4 Incrementally train model on new classes with synthetic training data

Building on PMI and feature distribution modeling, we explore the possibility of leveraging the rich knowledge encoded in CLIP to generate training data from text features via model inversion through the image encoder. This allows the model to learn new classes without requiring additional training data. To preserve generalization, we generate data using the original pre-trained CLIP model rather than its fine-tuned version, as fine-tuning may lead to overfitting and loss of pre-trained knowledge.

To obtain features for inversion, we propose semantic-aware feature projection. Specifically, we first compute a projection function that transforms text features of an old class $c$ to those of a new class $d$, and then apply this projection to image features of $c$ to produce pseudo image features for $d$. Since CLIP models normalize both image and text features onto a unit hypersphere, we use a rotation matrix as the projection function:

$$F_t(\mathbf{t}_d) = \mathbf{R}_{c,d} \cdot F_t(\mathbf{t}_c), \tag{7}$$

where $F_t$ denotes the text encoder of the CLIP model, $\mathbf{R}_{c,d}$ is the rotation matrix, $\mathbf{t}_c$ and $\mathbf{t}_d$ are hand-crafted prompts combined with the class names classes $c$ and $d$ respectively. In the data-free CL setting, where past data is unavailable, we model image feature distributions of previous classes and sample from them for mapping. The pseudo image feature of class $d$ is then obtained by

$$\mathbf{o}_{L,d} = \mathbf{R}_{c,d} \cdot \mathbf{o}_{L,c}, \tag{8}$$

Table 1: Final average accuracies on CIFAR-100 and Tiny-ImageNet with a ResNet-32 backbone. Numbers in parentheses indicate absolute improvements over the R-DFCIL baseline. Our method consistently outperforms existing baselines, with CFS providing additional gains.

| Method | Model Inversion | CIFAR-100 | | | Tiny-ImageNet | | |
|---|---|---|---|---|---|---|---|
| | | 5 task | 10 task | 20 task | 5 task | 10 task | 20 task |
| Upper bound | ✗ | 70.59 | 70.59 | 70.59 | 55.25 | 55.25 | 55.25 |
| DeepInversion | ✓ | 20.48 | 11.26 | 5.63 | - | - | - |
| ABD | ✓ | 48.84 | 36.75 | 24.40 | 30.83 | 23.17 | 14.61 |
| R-DFCIL | ✓ | 49.87 | 41.80 | 31.54 | 35.33 | 29.05 | 24.85 |
| **Ours w/o CFS** | ✓ | **52.05**(+2.18) | **43.23**(+1.43) | **32.23**(+0.69) | **37.65**(+2.32) | **32.09**(+3.04) | **25.51**(+0.66) |
| **Ours** | ✓ | **52.38**(+2.51) | **43.90**(+2.10) | **32.60**(+1.06) | **37.90**(+2.57) | **32.43**(+3.38) | **25.67**(+0.82) |

where $\mathbf{o}_{L,c}$ and $\mathbf{o}_{L,d}$ denotes the image features of classes $c$ and $d$ respectively. To prevent mapped features from drifting into other class regions—potentially degrading the quality of the generated data, we control feature variance by shifting them toward their corresponding text features by

$$\mathbf{o}'_{L,d} = ((1-\alpha)\mathbf{o}_{L,d} + \alpha F_t(\mathbf{t}_d)) / ||(1-\alpha)\mathbf{o}_{L,d} + \alpha F_t(\mathbf{t}_d)||_2, \qquad (9)$$

where $\mathbf{o}'_{L,d}$ denotes the adjusted feature used for model inversion.

## 4    Experiments

### 4.1    Continual learning results

We evaluate our method on both ResNet- and CLIP-based CL settings under the challenging class-incremental setting, where the model must classify all classes without access to task identity at inference. In all experiments presented in this section, synthetic samples are generated using our PMI+full-model inversion strategy. Detailed experimental settings are provided in Appendix I.

**Model inversion for ResNet-based CL.** To demonstrate the effectiveness and efficiency of our proposed method, we conduct continual learning experiments on CIFAR-100 [23] and Tiny-ImageNet [57], following the experimental setup of R-DFCIL [11] using a ResNet-32 [12] backbone. The classes from each dataset are evenly divided into 5, 10, and 20 disjoint tasks.

We compare our method against competitive data-free baselines, including DeepInversion [60], ABD [39], and R-DFCIL [11]. The variant *w/o CFS* refers to sampling features directly from the Gaussian distribution. We use 50 update steps for PMI and 160 steps for full-model inversion. Final average accuracy after training on all tasks is used as the evaluation metric, and results are averaged over multiple runs with different random seeds, as shown in Table 1.

Our method consistently outperforms existing data-free CL baselines across varying task lengths and datasets, demonstrating its effectiveness and robustness. Incorporating the contrastive model for feature selection further improves performance, highlighting the importance of feature consistency in model inversion and the effectiveness of CFS. A more comprehensive comparison, including non-inversion data-free baselines, is provided in Appendix D.1.

**Model inversion for CLIP-based CL.** To further demonstrate the efficiency and compatibility of our approach, we apply our model inversion method for CLIP-based CL settings on CIFAR-100, ImageNet-R [8], and CUB-200 [45]. For each dataset, all classes are evenly divided into 10 disjoint tasks. We follow the experimental setup of PROOF [67], and use the pre-trained CLIP model weights provided by OpenAI [34] for all experiments.

We integrate our inversion method with VPT [17], CODA-Prompt [40], and MoE-Adapter [61], generating 5 synthetic samples per class from previous tasks. For model inversion, we generate samples using the model trained on the previous task, applying 200 steps for PMI and 600 steps for full-model inversion. We also compare our method with rehearsal-based baselines, including iCaRL [35], MEMO [66], PROOF [67], and CLIP4CLIP [16]. Table 2 reports both the final average accuracy and the average accuracy across all incremental stages. Performance results for CLAP4CLP and AttriCLIP [50] are referred from [16]. Empirically, we found that generating output features from the first convolutional layer of the ViT backbone improves performance, and we adopt this approach for all inversion-related experiments.

Table 2: CL performance on CIFAR-100, ImageNet-R, and CUB-200. Numbers in parentheses indicate absolute improvements over the corresponding baselines. Our model inversion strategy consistently enhances the performance of all baseline methods across all datasets.

| Method | No Real Image Buffer | CIFAR-100 | | ImageNet-R | | CUB-200 | |
|---|---|---|---|---|---|---|---|
| | | Avg. | Last | Avg. | Last | Avg. | Last |
| iCaRL | ✗ | 82.20 | 64.15 | 72.69 | 54.65 | 82.39 | 75.11 |
| MEMO | ✗ | 85.31 | 75.24 | 80.39 | 74.19 | 77.72 | 65.95 |
| PROOF | ✗ | 86.95 | 79.32 | 85.53 | 80.16 | 85.13 | 79.69 |
| CLAP4CLIP | ✗ | 86.13 | 78.21 | 85.77 | 79.98 | 86.93 | 81.64 |
| ZS-CLIP | ✓ | 76.71 | 66.25 | 83.00 | 77.25 | 66.81 | 53.52 |
| AttriCLIP | ✓ | 79.31 | 68.45 | 83.09 | 76.53 | 65.26 | 52.12 |
| VPT | ✓ | 83.98 | 74.34 | 85.56 | 79.67 | 68.22 | 55.51 |
| CODA-P | ✓ | 83.64 | 75.80 | 81.92 | 74.37 | 73.05 | 61.83 |
| MoE-Adapter | ✓ | 87.29 | 79.40 | 86.67 | 82.00 | 73.34 | 61.07 |
| **Ours + VPT** | ✓ | **85.45**(+1.47) | **76.91**(+2.57) | **86.18**(+0.62) | **80.97**(+1.30) | **70.14**(+1.92) | **57.04**(+1.53) |
| **Ours + CODA-P** | ✓ | **84.78**(+1.14) | **76.23**(+0.43) | **84.58**(+2.66) | **76.93**(+2.56) | **74.37**(+1.32) | **64.63**(+2.80) |
| **Ours + MoE-Adapter** | ✓ | **88.35**(+1.06) | **81.06**(+1.66) | **87.90**(+1.23) | **83.03**(+1.03) | **78.98**(+5.64) | **67.26**(+6.19) |

Generating data using our method and replaying it consistently improves performance over data-free settings, demonstrating the effectiveness of our approach. Our method enhances both prompt-based and adapter-based continual learning methods, highlighting its broad compatibility. It also shows robustness across different datasets and baseline methods. Furthermore, the experiments demonstrate the scalability of our inversion technique to large pre-trained models beyond ResNets. Although our method underperforms rehearsal-based baselines on the fine-grained classification dataset CUB-200, it still provides substantial improvements over the non-rehearsal baselines. We also include CLIP-based CL experiments with LAION-400M pre-trained weights in Appendix D.2, which further demonstrate the effectiveness and robustness of our method.

**Finetuning CLIP model on new classes with synthetic training data.** To evaluate the effectiveness of our semantic-aware feature projection in generating samples for new classes, we conduct experiments on CIFAR-100 and ImageNet-R using MoE-Adapter [61], with training settings kept consistent with CL experiments in Table 2. For this setup, the training data for the final task is unavailable, and we generate 10 samples per new class via semantic-aware feature projection. As discussed in Section 3.4, these samples are generated using the original pre-trained CLIP model.

We report accuracies on old classes, new classes (classes of the last task), and all classes in Figure 2. The labels 'zs', 'w/o data', and 'w data' refer to evaluations on the original CLIP model (zero-shot), the model trained on the first 9 tasks (without data from the last task), and the model continually trained with generated data for the last task, respectively. After fine-tuning the CLIP model on the first 9 tasks, the model performs significantly better on the last task, indicating that it learns domain-specific knowledge from the training data. Furthermore, training with data generated from the original CLIP model leads to

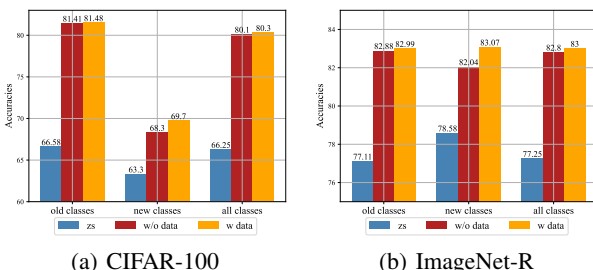

|   (a) CIFAR-100   |   (b) ImageNet-R   |

Figure 2: Accuracies on old, new, and all classes evaluated on the original CLIP model, the model after the 9th task, and the model continually trained on synthetic data. Finetuning the model with synthetic data further improves performance on both new and overall classes.

additional improvements on the last task and enhances overall accuracy across all tasks on both datasets, demonstrating the effectiveness of our method.

## 4.2 Ablation study

**Efficiency of applying PMI.** To demonstrate the efficiency of our PMI method, we plot the inversion loss curves during full-model inversion in Figure 3. The red curve represents input initialized using

PMI, while the orange curve uses randomly initialized inputs; all loss functions and optimization hyperparameters are kept identical. For the ViT backbone, we perform 200 update steps during PMI, while for the ResNet-32 backbone, we use 100 steps.

As shown in Figure 3, input samples initialized with PMI converge significantly faster than those with random initialization and achieves a much lower final loss. Even accounting for the additional steps used in PMI, our method reaches lower inversion loss with substantially fewer total steps. These results clearly demonstrate the efficiency of our PMI approach and showcase inputs optimized by PMI serves as good initialization for full-model inversion.

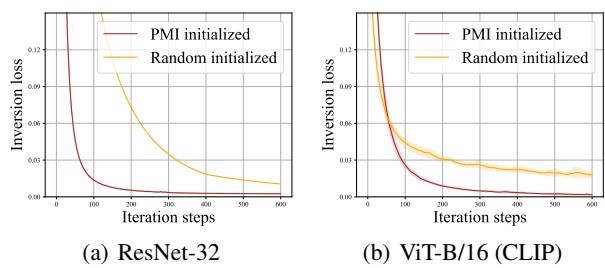

(a) ResNet-32    (b) ViT-B/16 (CLIP)

Figure 3: Inversion loss curves for PMI vs. random initialization. PMI converges faster and achieves lower final loss, confirming its efficiency.

**Compare with full-model inversion in CLIP-based CL.** To further demonstrate the efficiency of our PMI+full-model inversion strategy, we compare it with full-model inversion on CIFAR-100 in the CLIP-based CL. For full-model inversion, we follow the setting of [19], inputs are randomly initialized and are updated for 3400 steps, whereas our method requires only 600 steps. All other settings are kept identical.

As shown in Table 3, our method outperforms full-model inversion in CLIP-based CL while requiring significantly fewer iteration steps. This demonstrates that our inversion method can generate data with comparable information content to full-model inversion, while requiring over four times fewer update steps, demonstrating its effectiveness. These results also highlight the efficiency of our approach. Furthermore, we show that incorporating CFS also improves performance in the full-model inversion setting, demonstrating its robustness.

Table 3: CL performance using MoE-Adapter with model inversion on the CIFAR-100 dataset. CFS effectively boosts performance, and PMI initialization further improves results while requiring fewer update steps.

| PMI | CFS | Avg. Acc. | Final Acc. |
|-----|-----|-----------|------------|
| *No inversion* | | 87.29 | 79.40 |
| ✗ | ✗ | 87.82 | 79.57 |
| ✗ | ✓ | 88.13 | 80.53 |
| ✓ | ✓ | **88.35** | **81.06** |

We further compare the time cost of our PMI+full-model inversion strategy with full-model inversion and Sparse Model Inversion [14] in CLIP-based CL experiments. As shown in Appendix F.1, our method achieves better performance with lower time cost. In addition, we compare our PMI+full-model inversion strategy with generator-based methods on both ResNet-based and CLIP-based CL experiments (Appendix F.2). Compared to generator-based approaches, our method attains higher performance with comparable time cost in ResNet-based experiments, and achieves both higher performance and lower time cost in CLIP-based experiments. Further discussion on the advantages of our method over generator-based approaches is provided in Appendix F.2.

**Ablation study on CFS.** To evaluate the effectiveness of our CFS method, we conduct experiments using CODA-Prompt and MoE-Adapter, both with and without CFS. All experimental settings and evaluation metrics are kept consistent as Section 4.1. The results are presented in Table 4.

Applying CFS improves both final average accuracy and overall average accuracy across various methods and datasets, demonstrating its effectiveness and robustness in CLIP-based CL. These results also highlight the importance of maintaining feature distribution consistency in this setting.

**Visualization of generated samples.** To demonstrate that the inverted data captures key semantic features of each class, we visualize the generated samples for both old and new classes on CIFAR-100 and ImageNet-R, using our PMI+full-model inversion strategy. The first row shows samples from CIFAR-100, while the second row presents samples from ImageNet-R. For old classes, features are sampled from class-wise Gaussian distributions and selected using CFS, then used to generate images. For new classes, images are generated from features obtained via semantic-aware feature projection.

The visualization results in Figure 4 demonstrate that:**1)** features sampled from the modeled distributions and obtained via semantic-aware feature projection retain key semantic characteristics of

Table 4: CL performance comparison with and without CFS. Applying CFS consistently improves performance across both methods and all three datasets in nearly all cases.

| Method | CFS | CIFAR-100 | | ImageNet-R | | CUB 200 | |
|---|---|---|---|---|---|---|---|
| | | Avg. | Last | Avg. | Last | Avg. | Last |
| CODA-P+Inversion | ✗ | **84.93** | 76.04 | 84.32 | 76.63 | 73.32 | 63.91 |
| MoE-Adapter+Inversion | ✗ | 88.23 | 80.40 | 87.07 | 82.68 | 78.61 | 66.71 |
| CODA-P+Inversion | ✓ | 84.78 (-0.15) | **76.23** (+0.19) | **84.58** (+0.26) | **76.93** (+0.30) | **74.37** (+1.05) | **64.63** (+0.72) |
| MoE-Adapter+Inversion | ✓ | **88.35** (+0.12) | **81.06** (+0.66) | **87.90** (+0.83) | **83.03** (+0.35) | **78.98** (+0.37) | **67.26** (+0.55) |

the target class; and **2)** PMI+full model inversion strategy reliably reconstructs meaningful semantic content from the input features.

## 5 Conclusion

In this work, we address the inefficiency of model inversion by introducing PMI, which significantly reduces the number of iterations required for convergence. Additionally, we tackle the issue of feature distribution shift by modeling real feature distributions and sampling features from them for inversion. Building on PMI and feature distribution modeling, we demonstrate the feasibility of generating training samples for new classes via semantic-aware feature projection. Experimental results validate the effectiveness of our approach. Our model inversion method is also beneficial for other data-free tasks, such as knowledge distillation, meta-learning, and dataset distillation. Additionally, our model inversion method can be used as an efficient tool for interpreting and understanding what the model has learned.

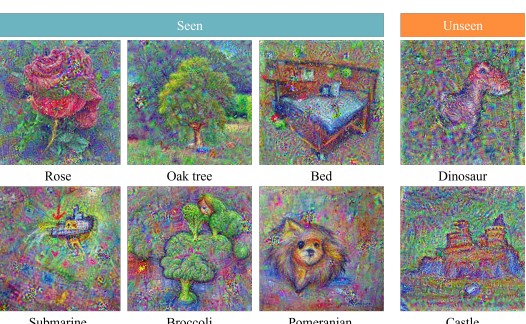

Figure 4: Visualization of generated samples for old and new classes from CIFAR-100 (top row) and ImageNet-R (bottom row). Our method reliably recovers the semantic information of each class.

**Limitations.** Our current approach for generating samples of new classes from CLIP models can be further enhanced by integrating more advanced techniques. For example, [32] uses class descriptions generated by large language models to improve CLIP's zero-shot performance. These descriptions could also be leveraged to guide sample generation for new classes. Wei et al. [55] enhance the diversity of synthetic data through style dictionary diversification while preserving consistency using classification objective during inversion. We leave this direction for future work.

Another limitation of our work lies in the loss of scale information when mapping the output of $f_{\text{cont}}$ onto a hypersphere. Uniformly projecting all selected features onto the hypersphere does not ensure that their distribution precisely matches that of real features, as the scale information between the two remains unaligned. In our approach, Gaussian distribution modeling and CFS provide a lightweight and efficient means of preserving feature distribution information. The consistent improvements achieved by CFS across different experimental settings support our insight that maintaining feature consistency enhances continual learning performance. More advanced distribution modeling techniques, such as VAEs [21] used in [9], could further improve performance.

## Acknowledgements

This paper was partially supported by Australian Research Council (ARC) Discovery Early Career Researcher Award (DECRA) project DE230101591 to D. Gong.

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

# Appendix

## A  Decomposing model inversion objective into layer-wise constraint

VMI [49] proposed a model inversion objective based on KL divergence and reformulated it into classification and prior terms. In addition to constraining the input prior distribution, DeepInversion [60] introduced a constraint on the prior distribution of intermediate layer outputs. Let $\mathbf{o}_l$ denote the output of the $l$-th layer, $L$ the total number of layers, $\mathbf{x}$ the input, and $y$ the label, we formulate our model inversion objective using KL divergence as:

$$p_s^*(\mathbf{x}) = \arg\min_{p_s(\mathbf{x})} D_{\mathrm{KL}}(p_s(\mathbf{o}_L, \mathbf{o}_{L-1}, \cdots, \mathbf{x}|y) || p_r(\mathbf{o}_L, \mathbf{o}_{L-1}, \cdots, \mathbf{x}|y)), \tag{10}$$

here $p_r(\cdot)$ denotes the distribution computed from real data, while $p_s(\cdot)$ represents the distribution from synthetic data. In the following, we show that this objective can be decomposed into a layer-wise inversion formulation.

Based on the chain rule of probabilities

$$\begin{aligned} p_s(\mathbf{o}_L, \mathbf{o}_{L-1}, \cdots, \mathbf{x}|y) &= p_s(\mathbf{o}_{L-1}, \cdots, \mathbf{x}|\mathbf{o}_L, y)p_s(\mathbf{o}_L|y), \\ p_r(\mathbf{o}_L, \mathbf{o}_{L-1}, \cdots, \mathbf{x}|y) &= p_r(\mathbf{o}_{L-1}, \cdots, \mathbf{x}|\mathbf{o}_L, y)p_r(\mathbf{o}_L|y), \end{aligned} \tag{11}$$

KL divergence in Eq. (10) can be decomposed by

$$\begin{aligned} &D_{\mathrm{KL}}(p_s(\mathbf{o}_L, \mathbf{o}_{L-1}, \cdots, \mathbf{x}|y) || p_r(\mathbf{o}_L, \mathbf{o}_{L-1}, \cdots, \mathbf{x}|y)) \\ =&\mathbb{E}_{p_s(\mathbf{o}_L, \mathbf{o}_{L-1}, \cdots, \mathbf{x}|y)} \ \log \frac{p_s(\mathbf{o}_L, \mathbf{o}_{L-1}, \cdots, \mathbf{x}|y)}{p_r(\mathbf{o}_L, \mathbf{o}_{L-1}, \cdots, \mathbf{x}|y)} \\ =&\mathbb{E}_{p_s(\mathbf{o}_{L-1}, \cdots, \mathbf{x}|\mathbf{o}_L, y)p_s(\mathbf{o}_L|y)} \ \log \frac{p_s(\mathbf{o}_{L-1}, \cdots, \mathbf{x}|\mathbf{o}_L, y)p_s(\mathbf{o}_L|y)}{p_r(\mathbf{o}_{L-1}, \cdots, \mathbf{x}|\mathbf{o}_L, y)p_r(\mathbf{o}_L|y)} \\ =&\int \cdots \int p_s(\mathbf{o}_{L-1}, \cdots, \mathbf{x}|\mathbf{o}_L, y)p_s(\mathbf{o}_L|y) \left( \log \frac{p_s(\mathbf{o}_{L-1}, \cdots, \mathbf{x}|\mathbf{o}_L, y)}{p_r(\mathbf{o}_{L-1}, \cdots, \mathbf{x}|\mathbf{o}_L, y)} + \log \frac{p_s(\mathbf{o}_L|y)}{p_r(\mathbf{o}_L|y)} \right) d\mathbf{x} \cdots d\mathbf{o}_L \\ =&\int p_s(\mathbf{o}_L|y) \left( \int \cdots \int p_s(\mathbf{o}_{L-1}, \cdots, \mathbf{x}|\mathbf{o}_L, y) \ \log \frac{p_s(\mathbf{o}_{L-1}, \cdots, \mathbf{x}|\mathbf{o}_L, y)}{p_r(\mathbf{o}_{L-1}, \cdots, \mathbf{x}|\mathbf{o}_L, y)} d\mathbf{x} \cdots d\mathbf{o}_{L-1} \right) d\mathbf{o}_L \\ &+ \int \cdots \int p_s(\mathbf{o}_{L-1}, \cdots, \mathbf{x}|\mathbf{o}_L, y) \left( \int p_s(\mathbf{o}_L|y) \ \log \frac{p_s(\mathbf{o}_L|y)}{p_r(\mathbf{o}_L|y)} d\mathbf{o}_L \right) d\mathbf{x} \cdots d\mathbf{o}_{L-1} \\ =&\mathbb{E}_{p_s(\mathbf{o}_L|y)} D_{\mathrm{KL}}(p_s(\mathbf{o}_{L-1}, \cdots, \mathbf{x}|\mathbf{o}_L, y) || p_r(\mathbf{o}_{L-1}, \cdots, \mathbf{x}|\mathbf{o}_L, y)) \\ &+ \mathbb{E}_{p_s(\mathbf{o}_{L-1}, \cdots, \mathbf{x}|\mathbf{o}_L, y)} D_{\mathrm{KL}}(p_s(\mathbf{o}_L|y) || p_r(\mathbf{o}_L|y)). \end{aligned} \tag{12}$$

The KL divergence in the second term of Eq. (12) is not included in the integration over $\mathbf{o}_{L-1}, \cdots, \mathbf{x}$, and the expectation simplifies to the KL divergence itself: $\mathbb{E}_{p_s(\mathbf{o}_{L-1}, \cdots, \mathbf{x}|\mathbf{o}_L, y)} D_{\mathrm{KL}}(p_s(\mathbf{o}_L|y) || p_r(\mathbf{o}_L|y)) = D_{\mathrm{KL}}(p_s(\mathbf{o}_L|y) || p_r(\mathbf{o}_L|y))$. We further assume that inversion through each layer retains essential information. Specifically, $\mathbf{o}_L$ is assumed to contain all information about the label $y$, and the representations $\mathbf{x}, \cdots \mathbf{o}_{L-1}$ can be recovered from $\mathbf{o}_L$. This assumption implies a conditional independence relationship, which can be formulated as:

$$p_s(\mathbf{o}_{L-1}, \cdots, \mathbf{x}|\mathbf{o}_L, y) \approx p_s(\mathbf{o}_{L-1}, \cdots, \mathbf{x}|\mathbf{o}_L), \quad p_r(\mathbf{o}_{L-1}, \cdots, \mathbf{x}|\mathbf{o}_L, y) \approx p_r(\mathbf{o}_{L-1}, \cdots, \mathbf{x}|\mathbf{o}_L). \tag{13}$$

Therefore, the decomposed objective is:

$$\begin{aligned} &D_{\mathrm{KL}}(p_s(\mathbf{o}_L, \mathbf{o}_{L-1}, \cdots, \mathbf{x}|y) || p_r(\mathbf{o}_L, \mathbf{o}_{L-1}, \cdots, \mathbf{x}|y)) \\ =&\mathbb{E}_{p_s(\mathbf{o}_L|y)} D_{\mathrm{KL}}(p_s(\mathbf{o}_{L-1}, \cdots, \mathbf{x}|\mathbf{o}_L, y) || p_r(\mathbf{o}_{L-1}, \cdots, \mathbf{x}|\mathbf{o}_L, y)) \\ &+ \mathbb{E}_{p_s(\mathbf{o}_{L-1}, \cdots, \mathbf{x}|\mathbf{o}_L, y)} D_{\mathrm{KL}}(p_s(\mathbf{o}_L|y) || p_r(\mathbf{o}_L|y)) \\ \approx&\mathbb{E}_{p_s(\mathbf{o}_L|y)} D_{\mathrm{KL}}(p_s(\mathbf{o}_{L-1}, \cdots, \mathbf{x}|\mathbf{o}_L) || p_r(\mathbf{o}_{L-1}, \cdots, \mathbf{x}|\mathbf{o}_L)) + D_{\mathrm{KL}}(p_s(\mathbf{o}_L|y) || p_r(\mathbf{o}_L|y)). \end{aligned} \tag{14}$$

The expectation on KL divergence of the right hand side can be decomposed with the same method as:

$$\mathbb{E}_{p_s(\mathbf{o}_L|y)}\left[D_{\mathrm{KL}}(p_s(\mathbf{o}_{L-1},\cdots,\mathbf{x}|\mathbf{o}_L)||p_r(\mathbf{o}_{L-1},\cdots,\mathbf{x}|\mathbf{o}_L))\right]$$

$$\approx\mathbb{E}_{p_s(\mathbf{o}_L|y)}\left[\mathbb{E}_{p_s(\mathbf{o}_{L-1}|\mathbf{o}_L)}D_{\mathrm{KL}}(p_s(\mathbf{o}_{L-2},\cdots,\mathbf{x}|\mathbf{o}_{L-1})||p_r(\mathbf{o}_{L-2},\cdots,\mathbf{x}|\mathbf{o}_{L-1}))\right]$$

$$+\mathbb{E}_{p_s(\mathbf{o}_L|y)}D_{\mathrm{KL}}(p_s(\mathbf{o}_{L-1}|\mathbf{o}_L)||p_r(\mathbf{o}_{L-1}|\mathbf{o}_L)) \tag{15}$$

$$\approx\mathbb{E}_{p_s(\mathbf{o}_{L-1}|y)}\left[D_{\mathrm{KL}}(p_s(\mathbf{o}_{L-2},\cdots,\mathbf{x}|\mathbf{o}_{L-1})||p_r(\mathbf{o}_{L-2},\cdots,\mathbf{x}|\mathbf{o}_{L-1}))\right]$$

$$+\mathbb{E}_{p_s(\mathbf{o}_L|y)}D_{\mathrm{KL}}(p_s(\mathbf{o}_{L-1}|\mathbf{o}_L)||p_r(\mathbf{o}_{L-1}|\mathbf{o}_L)).$$

The last line in Eq. (15) is based on conditional independence, namely

$$p_s(\mathbf{o}_{L-1}|\mathbf{o}_L,y)\approx p_s(\mathbf{o}_{L-1}|\mathbf{o}_L)$$

$$p_s(\mathbf{o}_{L-1}|,y)=\int p_s(\mathbf{o}_{L-1}|\mathbf{o}_L,y)p_s(\mathbf{o}_L|y)d\mathbf{o}_L \tag{16}$$

$$\approx\int p_s(\mathbf{o}_{L-1}|\mathbf{o}_L)p_s(\mathbf{o}_L|y)d\mathbf{o}_L.$$

The first term in the right hand side of Eq. (15) can be decomposed in the same way as:

$$\mathbb{E}_{p_s(\mathbf{o}_{L-1}|y)}\left[D_{\mathrm{KL}}(p_s(\mathbf{o}_{L-2},\cdots,\mathbf{x}|\mathbf{o}_{L-1})||p_r(\mathbf{o}_{L-2},\cdots,\mathbf{x}|\mathbf{o}_{L-1}))\right]$$

$$\approx\mathbb{E}_{p_s(\mathbf{o}_{L-1}|y)}\left[\mathbb{E}_{p_s(\mathbf{o}_{L-2}|\mathbf{o}_{L-1})}D_{\mathrm{KL}}(p_s(\mathbf{o}_{L-3},\cdots,\mathbf{x}|\mathbf{o}_{L-2})||p_r(\mathbf{o}_{L-3},\cdots,\mathbf{x}|\mathbf{o}_{L-2}))\right]$$

$$+\mathbb{E}_{p_s(\mathbf{o}_{L-1}|y)}D_{\mathrm{KL}}(p_s(\mathbf{o}_{L-2}|\mathbf{o}_{L-1})||p_r(\mathbf{o}_{L-2}|\mathbf{o}_{L-1})) \tag{17}$$

$$\approx\mathbb{E}_{p_s(\mathbf{o}_{L-2}|y)}\left[D_{\mathrm{KL}}(p_s(\mathbf{o}_{L-3},\cdots,\mathbf{x}|\mathbf{o}_{L-2})||p_r(\mathbf{o}_{L-3},\cdots,\mathbf{x}|\mathbf{o}_{L-2}))\right]$$

$$+\mathbb{E}_{p_s(\mathbf{o}_{L-1}|y)}D_{\mathrm{KL}}(p_s(\mathbf{o}_{L-2}|\mathbf{o}_{L-1})||p_r(\mathbf{o}_{L-2}|\mathbf{o}_{L-1})).$$

Given $\mathbf{o}_0=\mathbf{x}$ and $\mathbf{o}_{L+1}=y$, decomposing inversion objective layer-by-layer recursively results in

$$D_{\mathrm{KL}}(p_s(\mathbf{o}_L,\mathbf{o}_{L-1},\cdots,\mathbf{x}|y)||p_r(\mathbf{o}_L,\mathbf{o}_{L-1},\cdots,\mathbf{x}|y))$$

$$\approx\sum_{l=1}^{L}\mathbb{E}_{p_s(\mathbf{o}_{l+1}|y)}\left[\mathbb{E}_{p_s(\mathbf{o}_l|\mathbf{o}_{l+1})}D_{\mathrm{KL}}(p_s(\mathbf{o}_{l-1}|\mathbf{o}_l)||p_r(\mathbf{o}_{l-1}|\mathbf{o}_l))\right]+D_{\mathrm{KL}}(p_s(\mathbf{o}_L|y)||p_r(\mathbf{o}_L|y)). \tag{18}$$

Eq. (18) shows that, under the conditional independence assumption, the input to each layer can be inferred from its output. Summing the layer-wise KL divergences thus provides an approximation of the total inversion objective in Eq. (10).

## B  Layer-wise and full-model inversion objective

The KL divergences in Eq. (18) are intractable in practice. Following [49], we reformulate them using Bayes' rule as:

$$D_{\mathrm{KL}}(p_s(\mathbf{o}_{l-1}|\mathbf{o}_l)||p_r(\mathbf{o}_{l-1}|\mathbf{o}_l))$$

$$=\mathbb{E}_{p_s(\mathbf{o}_{l-1}|\mathbf{o}_l)}\log\frac{p_s(\mathbf{o}_{l-1}|\mathbf{o}_l)}{p_r(\mathbf{o}_{l-1}|\mathbf{o}_l)}$$

$$=\mathbb{E}_{p_s(\mathbf{o}_{l-1}|\mathbf{o}_l)}\log\frac{p_s(\mathbf{o}_{l-1}|\mathbf{o}_l)p_r(\mathbf{o}_l)}{p_r(\mathbf{o}_l|\mathbf{o}_{l-1})p_r(\mathbf{o}_{l-1})} \tag{19}$$

$$=\mathbb{E}_{p_s(\mathbf{o}_{l-1}|\mathbf{o}_l)}\log\frac{p_s(\mathbf{o}_{l-1}|\mathbf{o}_l)}{p_r(\mathbf{o}_{l-1})}-\mathbb{E}_{p_s(\mathbf{o}_{l-1}|\mathbf{o}_l)}\log\,p_r(\mathbf{o}_l|\mathbf{o}_{l-1})+\log\,p_r(\mathbf{o}_l)$$

$$=D_{\mathrm{KL}}(p_s(\mathbf{o}_{l-1}|\mathbf{o}_l)||p_r(\mathbf{o}_{l-1}))-\mathbb{E}_{p_s(\mathbf{o}_{l-1}|\mathbf{o}_l)}\log\,p_r(\mathbf{o}_l|\mathbf{o}_{l-1})+\log\,p_r(\mathbf{o}_l).$$

Since $p_r(\mathbf{o}_l)$ is computed from real data and is independent of the synthetic data, we treat it as a constant. The negative log-probability term $-\log\,p_r(\mathbf{o}_l|\mathbf{o}_{l-1})$ is approximated using mean squared error (MSE). For the KL divergence at the final layer $D_{\mathrm{KL}}(p_s(\mathbf{o}_L|y)||p_r(\mathbf{o}_L|y))$, the negative log-probability $-\log\,p_r(y|\mathbf{o}_l)$ can be approximated using cross-entropy (CE) loss for classification tasks, and mean squared error (MSE) for regression tasks. In this work, we adopt the CE loss to construct our final inversion objective. For the prior constraint $D_{\mathrm{KL}}(p_s(\mathbf{o}_{l-1}|\mathbf{o}_l)||p_r(\mathbf{o}_{l-1}))$, we follow ABD [39] by approximating both distributions as Gaussians and computing the KL divergence in closed form.

The expectation $\mathbb{E}_{p_s(\mathbf{o}_{l+1}|y)}\left[\mathbb{E}_{p_s(\mathbf{o}_l|\mathbf{o}_{l+1})}D_{\mathrm{KL}}(p_s(\mathbf{o}_{l-1}|\mathbf{o}_l)||p_r(\mathbf{o}_{l-1}|\mathbf{o}_l))\right]$ can be approximated by averaging the KL divergences computed over a batch of $\mathbf{o}_l$ if $\mathbf{o}_l$ is known. Building on the layer-wise constraint in Eq. (18) and the approximations above, $\mathbf{o}_L$ can be optimized with given $y$ using objective

$$D_{\mathrm{KL}}(p_s(\mathbf{o}_L|y)||p_r(\mathbf{o}_L|y))$$
$$\approx D_{\mathrm{KL}}(\mathcal{N}(\hat{\boldsymbol{\mu}}_L,\hat{\boldsymbol{\sigma}}_L)||\mathcal{N}(\boldsymbol{\mu}_L,\boldsymbol{\sigma}_L)) + \frac{1}{N}\sum_{i=1}^{N}\ell_{\mathrm{CE}}(\mathbf{o}_{L,i},y;\boldsymbol{\theta}_{L+1}). \tag{20}$$

Therefore, we propose a top-down optimization strategy that optimize features layer-by-layer from the output layer to the input. At each step, the optimized $\mathbf{o}_{l+1}$ is used as the target output to guide the optimization of $\mathbf{o}_l$. In other words, $\mathbf{o}_l$ is optimized before proceeding to $\mathbf{o}_{l-1}$. The layer-wise inversion objective is

$$\mathbb{E}_{p_s(\mathbf{o}_{l+1}|y)}\left[\mathbb{E}_{p_s(\mathbf{o}_l|\mathbf{o}_{l+1})}D_{\mathrm{KL}}(p_s(\mathbf{o}_{l-1}|\mathbf{o}_l)||p_r(\mathbf{o}_{l-1}|\mathbf{o}_l))\right]$$
$$\approx D_{\mathrm{KL}}(\mathcal{N}(\hat{\boldsymbol{\mu}}_{l-1},\hat{\boldsymbol{\sigma}}_{l-1})||\mathcal{N}(\boldsymbol{\mu}_{l-1},\boldsymbol{\sigma}_{l-1})) + \frac{1}{N}\sum_{i=1}^{N}\ell_{\mathrm{MSE}}(\mathbf{o}_{l-1,i},\mathbf{o}_{l,i};\boldsymbol{\theta}_l), \tag{21}$$

where $\boldsymbol{\theta}_l$ denotes the parameters of the $l$-th layer, $\mu_{l-1}$ and $\sigma_{l-1}$ are the mean and standard deviation computed from real data, and $\hat{\mu}_{l-1}$ and $\hat{\sigma}_{l-1}$ are the corresponding statistics computed from synthetic data. The model inversion objective over all layers is given as:

$$D_{\mathrm{KL}}(p_s(\mathbf{o}_L,\mathbf{o}_{L-1},\cdots,x|y)||p_r(\mathbf{o}_L,\mathbf{o}_{L-1},\cdots,x|y))$$
$$\approx \sum_{l=1}^{L}\left(D_{\mathrm{KL}}(\mathcal{N}(\hat{\boldsymbol{\mu}}_{l-1},\hat{\boldsymbol{\sigma}}_{l-1})||\mathcal{N}(\boldsymbol{\mu}_{l-1},\boldsymbol{\sigma}_{l-1})) + \frac{1}{N}\sum_{i=1}^{N}\ell_{\mathrm{MSE}}(\mathbf{o}_{l-1,i},\mathbf{o}_{l,i};\boldsymbol{\theta}_l)\right)$$
$$+ D_{\mathrm{KL}}(\mathcal{N}(\hat{\boldsymbol{\mu}}_L,\hat{\boldsymbol{\sigma}}_L)||\mathcal{N}(\boldsymbol{\mu}_L,\boldsymbol{\sigma}_L)) + \frac{1}{N}\sum_{i=1}^{N}\ell_{\mathrm{CE}}(\mathbf{o}_{L,i},y;\boldsymbol{\theta}_{L+1}). \tag{22}$$

Performing model inversion on the full model omits the loss terms $\ell_{\mathrm{MSE}}(\mathbf{o}_{l-1,i},\mathbf{o}_{l,i};\boldsymbol{\theta}_l)$, as there is no available target $\mathbf{o}_{l,i}$ for the output of the $l$-th layer. Equation (22) is equivalent to the model inversion objective proposed in ABD [39], excluding the smoothness constraint. Since the loss landscape of a single layer is much simpler than that of the full model, inversion at the layer level requires significantly fewer update steps.

## C  Detailed algorithms

### C.1  Implementation details on ResNets-based CL

Since ResNets are trained from scratch in CL, the feature representations of previous classes may shift after learning new tasks. To account for this, we update the mean and standard deviation of previous classes using synthetic data after each task. The contrastive models for each previous class are then retrained accordingly. Once the class-wise statistics and contrastive models are updated, we sample features for model inversion.

For CL with ResNets, we enhance the separability of features from new tasks by incorporating classification layer fine-tuning, following ABD [39], and based on the replay loss from R-DFCIL [11]. The loss function used during training on task $t$ is:

$$\mathcal{L}_{\mathrm{CL}}(\boldsymbol{\theta}_t) = \frac{1}{N}\sum_{i=1}^{N}\ell_{\mathrm{lCE}}(\mathbf{x}_i,y_i;\boldsymbol{\theta}_t) + \frac{1}{M}\sum_{j=1}^{M}(\lambda_{\mathrm{hkd}}\mathcal{L}_{\mathrm{hkd}}(\mathbf{x}_j;\boldsymbol{\theta}_t,\boldsymbol{\theta}_{t-1}) + \lambda_{\mathrm{rkd}}\mathcal{L}_{\mathrm{rkd}}(\mathbf{x}_j;))$$
$$+ \lambda_{\mathrm{ft}}\frac{1}{M+N}\sum_{k=1}^{M+N}\mathcal{L}_{\mathrm{ft}}(\mathbf{x}_k,y_k;\boldsymbol{\theta}_t), \tag{23}$$

where $\ell_{\mathrm{lCE}}$ denotes the local cross-entropy loss, computed only over the classes of the current task. $\mathcal{L}_{\mathrm{ft}}$, proposed in [39], is a cross-entropy loss over all old and current classes, and is used exclusively to update the classification layer. $\mathcal{L}_{\mathrm{hkd}}$ and $\mathcal{L}_{\mathrm{rkd}}$ refer to the hard knowledge distillation (HKD) loss

and the relational knowledge distillation (RKD) loss, respectively, both introduced in [11]. $\boldsymbol{\theta}_t$ denotes the parameters of the CL model at task $t$, and $\boldsymbol{\theta}_{t-1}$ represents the parameters after training on the previous task. $\lambda_{\text{hkd}}$, $\lambda_{\text{rkd}}$, and $\lambda_{\text{ft}}$ are hyperparameters that weight their corresponding loss terms to balance their contributions during optimization, and we apply the loss factor change scheme proposed in R-DFCIL. Detailed algorithm is presented in Alg. 3, and analysis on the effect of $\mathcal{L}_{\text{ft}}$ is provided in Appendix D.1.

---

**Algorithm 3:** ResNet-Based CL

---

**Input:** Dataset sequence $\mathcal{D}_{1:T}$
**Output:** Trained CL model parameters $\boldsymbol{\theta}_T$
Initialize model parameter $\boldsymbol{\theta}_0$;
**for** $t = 1$ **to** $T$ **do**
    **if** $i > 1$ **then**
        Sample features for inversion;
        PMI+full-model inversion to generate previous data;
        Train model with replay;
    **else**
        Train model with CE loss;
    Finetune classification layer;
    Compute class-wise feature Gaussian distribution for new classes;
    Update class-wise feature Gaussian distribution for old classes;
    Train contrastive model for each old class;
    Save model as teacher model;

---

## C.2 Implementation details on CLIP-based CL

Since CLIP models encode rich pre-trained knowledge, their image features already contain strong semantic information for classification. We therefore assume that the features of previously learned classes do not drift significantly. As a result, we do not update the feature statistics or contrastive models for previous classes in CLIP-based CL. For all our methods, classification is performed using local cross-entropy loss, computed only on the current task classes.

We integrate our data generation method into three baseline approaches: VPT [17], CODA-Prompt [40], and MoE-Adapter [61]. VPT introduces learnable prompts into the vision encoder while keeping the text encoder frozen. CODA-Prompt extends the learnable prompts in the image encoder after each task and uses a learnable classification head. For both methods, we apply only the hard knowledge distillation (HKD) loss on synthetic data during replay to mitigate forgetting. MoE-Adapter incorporates a Mixture of Experts (MoE) into both the image and text encoders. To prevent forgetting on text encoder, we introduce a text knowledge distillation loss to prevent forgetting in the text encoder by

---

**Algorithm 4:** CLIP-based CL

---

**Input:** CLIP model $F_{t,0}$, $F_{i,0}$, task sequence dataset $\mathcal{D}_{1:T}$
**Output:** Finetuned CLIP model $F_{t,T}$, $F_{i,T}$
**for** $k = 1$ **to** $T$ **do**
    **if** $k > 1$ **then**
        Sample features for inversion;
        PMI+full-model inversion to generate previous data;
        Train CLIP model with replay;
    **else**
        Train model with CE loss;
    Compute class-wise feature Gaussian distribution;
    Train contrastive model for each new class;
    Update layer-wise input distribution;
    Save model as teacher model;

---

$$\mathcal{L}_{\text{tkd}}(\mathbf{t}_c; F_{t,k-1}, F_{t,k}) = ||F_{t,k-1}(\mathbf{t}_c) - F_{t,k}(\mathbf{t}_c)||_1, \tag{24}$$

where $F_{t,k}$ denotes the text encoder during training on the $k$-th task, and $F_{t,k-1}$ refers to the text encoder after training on the previous task. Additionally, we introduce a text encoder fine-tuning loss to improve classification performance over all old classes, defined as:

$$\mathcal{L}_{\text{tft}}(\mathbf{x}_c, \mathbf{t}_c; F_i, F_t) = \mathcal{L}_{CE}(F_i(\mathbf{x}_c), F_t(\mathbf{t}_c)). \tag{25}$$

The text encoder fine-tuning loss is used exclusively to update the text encoder for improved classification performance over all old classes. Loss factor changing scheme proposed in R-DFCIL is also applied in our CLIP-based CL method. Detailed algorithm is shown in Alg. 4.

In practice, we found that performing model inversion through the first convolutional layer of the ViT model significantly degrades CL performance. This is because the layer uses a kernel size of 16 and a stride of 16, leading to substantial information loss from the input. Moreover, this layer remains frozen during CL. To address this, we implement model inversion to generate the output features of the first convolutional layer and apply this approach across all CLIP-based CL experiments.

### C.3 Model inversion details

Based on the model inversion loss in Eq. (27), prior works [19, 60, 39] introduce a smoothness constraint on the synthetic data. In our proposed PMI method, we omit this constraint, as enforcing smoothness may result in the loss of important information in the feature maps.

For full-model inversion, we find that removing the smoothness constraint does not affect CL performance in ResNet-based experiments. Therefore, we omit it in this setting. In CLIP-based CL experiments, we perform model inversion to generate the output feature map of the first convolutional layer. To emulate the smoothness constraint typically applied to input images, we apply a total variation loss to this feature map, with a fixed weight of $5 \times 10^{-3}$ for all CL experiments. For visualization experiments, we follow the same setup and apply total variation loss with the same weight, consistent with the setting in [19].

The prior distribution constraint in Eq. (21) requires layer-wise statistics of real data. Unlike ResNets, the ViT backbone used in CLIP does not include batch normalization layers. To apply the prior distribution constraint to the CLIP model, we compute the mean and standard deviation over real data after training each task and maintain a moving average across all previously seen tasks. In practice, we treat each residual block as a single layer in ResNet architectures, and each residual transformer block as a single layer in the ViT backbone of the CLIP model.

In practice, we introduce a scaling factor for each loss term in both the PMI objective and the full-model inversion objective to balance their relative contributions. Specifically, the layer-wise inversion objective is given by:

$$\mathcal{L}_{\text{inv}}(\mathbf{o}_{l-1}; \boldsymbol{\theta}_l) = \alpha_l D_{\text{KL}}(\mathcal{N}(\hat{\boldsymbol{\mu}}_{l-1}, \hat{\boldsymbol{\sigma}}_{l-1}) || \mathcal{N}(\boldsymbol{\mu}_{l-1}, \boldsymbol{\sigma}_{l-1})) + \beta_l \frac{1}{N} \sum_{i=1}^{N} \ell_{\text{MSE}}(\mathbf{o}_{l-1,i}, \mathbf{o}_{l,i}; \boldsymbol{\theta}_l), \quad (26)$$

where $\alpha_l$ and $\beta_l$ are the scaling factors for the prior distribution constraint and the output constraint at layer $l$, respectively. The overall model inversion loss across all layers is then defined as:

$$\mathcal{L}_{\text{inv}}(\mathbf{x}; \boldsymbol{\theta}) = \sum_{l=1}^{L} \left( \alpha_l D_{\text{KL}}(\mathcal{N}(\hat{\boldsymbol{\mu}}_{l-1}, \hat{\boldsymbol{\sigma}}_{l-1}) || \mathcal{N}(\boldsymbol{\mu}_{l-1}, \boldsymbol{\sigma}_{l-1})) + \beta_l \frac{1}{N} \sum_{i=1}^{N} \ell_{\text{MSE}}(\mathbf{o}_{l-1,i}, \mathbf{o}_{l,i}; \boldsymbol{\theta}_l) \right)$$
$$+ \alpha_{L+1} D_{\text{KL}}(\mathcal{N}(\hat{\boldsymbol{\mu}}_L, \hat{\boldsymbol{\sigma}}_L) || \mathcal{N}(\boldsymbol{\mu}_L, \boldsymbol{\sigma}_L)) + \beta_{L+1} \frac{1}{N} \sum_{i=1}^{N} \ell_{\text{CE}}(\mathbf{o}_{L,i}, y; \boldsymbol{\theta}_{L+1}) + \gamma \mathcal{L}_{\text{tv}}(\mathbf{x}),$$
$$(27)$$

where $\mathcal{L}_{\text{tv}}$ denotes the total variation loss used to enforce smoothness, $\gamma$ is its corresponding weighting factor, and $\boldsymbol{\theta}_l$ represents the parameters of the $l$-th layer.

## D  Additional experiment results

### D.1  ResNet-based CL

To further demonstrate the effectiveness of our method, we include DCMI [33] as well as data-free methods SSRE [69] and PRAKA [38], which do not rely on model inversion. The final average accuracy is reported in Table 5. To specifically evaluate the impact of our PMI + full-model inversion strategy and feature modeling approach, we additionally incorporate the classification head fine-tuning loss into R-DFCIL and compare the results with our method in terms of final average accuracy on CIFAR-100 dataset, as shown in Table 6. The variant *R-DFCIL+* denotes R-DFCIL with the

Table 5: Final average accuracies on CIFAR-100 and Tiny-ImageNet using a ResNet-32 backbone, including the DCMI baseline. Red and blue values indicate the best and second-best performance, respectively. Our method consistently outperforms all existing baselines across all settings.

| Method | Model inversion | CIFAR-100 | | | Tiny-ImageNet | | |
|---|---|---|---|---|---|---|---|
| | | 5 task | 10 task | 20 task | 5 task | 10 task | 20 task |
| Upper bound | ✗ | 70.59±0.14 | 70.59±0.14 | 70.59±0.14 | 55.25±0.41 | 55.25±0.41 | 55.25±0.41 |
| SSRE | ✗ | 30.39±0.04 | 17.77±0.14 | 10.97±0.48 | 26.22±0.32 | 18.58±0.43 | 10.09±0.34 |
| PRAKA | ✗ | 37.74±0.48 | 26.72±0.38 | 16.32±0.66 | 31.69±0.20 | 22.37±0.14 | 13.62±0.42 |
| DeepInversion | ✓ | 20.48±1.11 | 11.26±0.46 | 5.63±0.10 | - | - | - |
| ABD | ✓ | 48.84±0.33 | 36.75±0.45 | 24.40±0.60 | 30.83±0.46 | 23.17±0.45 | 14.61±0.47 |
| DCMI | ✓ | 41.05±0.67 | 27.70±1.07 | 18.09±0.85 | 35.78±0.37 | 25.89±0.17 | 17.03±0.08 |
| R-DFCIL | ✓ | 49.87±0.45 | 41.80±0.24 | 31.54±0.54 | 35.33±0.02 | 29.05±0.28 | 24.85±0.16 |
| **Ours w/o CFS** | ✓ | **52.05±0.02** | **43.23±0.28** | **32.23±0.42** | **37.65±0.24** | **32.09±0.20** | **25.51±0.56** |
| **Ours** | ✓ | **52.38±0.53** | **43.90±0.35** | **32.60±0.29** | **37.90±0.10** | **32.43±0.09** | **25.67±0.71** |

Table 7: CLIP performance on CIFAR-100 and ImageNet-R using a CLIP model pre-trained on LAION-400M. Numbers in parentheses indicate the absolute improvement over the corresponding baseline methods. Our method consistently enhances the performance of all baselines on both datasets.

| Method | No Real Image Buffer | CIFAR-100 | | ImageNet-R | |
|---|---|---|---|---|---|
| | | Avg. | Last | Avg. | Last |
| iCaRL | ✗ | 79.91 | 63.94 | 72.22 | 54.38 |
| MEMO | ✗ | 84.67 | 74.98 | 80.00 | 74.07 |
| PROOF | ✗ | 86.70 | 79.05 | 85.34 | 80.10 |
| ZS-CLIP | ✓ | 81.38 | 71.26 | 82.93 | 76.67 |
| VPT | ✓ | 84.81 | 74.75 | 84.99 | 79.45 |
| CODA-P | ✓ | 85.00 | 76.56 | 83.70 | 75.73 |
| MoE-Adapter | ✓ | 88.01 | 79.97 | 85.75 | 80.83 |
| **Ours + VPT** | ✓ | **86.24** (1.43) | **76.13** (1.38) | **85.87** (0.88) | **80.18** (0.73) |
| **Ours + CODA-P** | ✓ | **85.84** (0.84) | **77.66** (1.10) | **84.82** (1.12) | **76.15** (0.42) |
| **Ours + MoE-Adapter** | ✓ | **88.55** (0.54) | **80.51** (0.54) | **87.18** (1.43) | **82.48** (1.65) |

classification layer fine-tuning loss, i.e., the loss function is identical to that used in our method. All experiments follow the same setup as described in Section 4.1.

As shown in Table 5, our method continues to outperform other baselines even after including DCMI in the comparison, further demonstrating its effectiveness. The non-inversion baselines, SSRE and PRAKA, generally underperform compared to the inversion-based methods ABD, DCMI, R-DFCIL, and our approach, as they rely heavily on the knowledge learned from the first task. These results highlight the effectiveness and robustness of model inversion in mitigating forgetting. In Table 6, incorporating the classification head fine-tuning loss slightly improves the performance of R-DFCIL; however, our method still surpasses the *R-DFCIL+* variant. This suggests that the performance gain primarily stems from our PMI+full-model inversion strategy and the proposed CFS method. These results provide strong evidence of the effectiveness of our approach.

## D.2 CLIP-based CL with LAION-400M pre-trained weight

To further demonstrate the robustness of our method, we conduct additional experiments using CLIP models pre-trained on LAION-400M, evaluated on CIFAR-100 and ImageNet-R. Our method is integrated with VPT [17], CODA-Prompt [40], and MoE-Adapter [61]. All hyperparameters are kept consistent with those used in Section 4.1. The final average accuracy and the average accuracy across all incremental stages are reported in Table 7. Our method consistently improves the performance of all baseline methods on both datasets, further demonstrating its effectiveness, robustness, and compatibility.

Table 8: Performance of MoE-Adapter combined with our method under different temperatures in the contrastive loss. The results show that performance remains stable across temperature values.

| Temperature | Avg. | Last |
|:-----------:|:----:|:----:|
| 0.5 | 88.26 | 80.96 |
| 0.8 | 88.27 | 81.16 |
| 1.0 | 88.35 | 81.06 |
| 1.5 | 88.24 | 80.94 |

## D.3  Consideration on temperature in contrastive loss

Following prior studies [51, 46], the temperature parameter $\tau$ is used to control the sharpness of the similarity distribution in the contrastive loss. To examine its effect in the negative contrastive loss, we incorporate temperature as shown in Eq. (28). A smaller $\tau$ amplifies gradients for hard negatives, encouraging fine-grained discrimination, whereas a larger $\tau$ smooths the

Table 6: Ablation study on classification head finetuning loss on CIFAR-100 dataset.

| Method | 5 task | 10 task | 20 task |
|--------|--------|---------|---------|
| R-DFCIL | 49.87±0.45 | 41.80±0.24 | 31.54±0.54 |
| R-DFCIL+ | 50.45±0.21 | 41.93±0.56 | 31.19±0.26 |
| **Ours w/o CFS** | **52.05±0.02** | **43.23±0.28** | **32.23±0.42** |
| **Ours** | **52.38±0.53** | **43.90±0.35** | **32.60±0.29** |

weighting across samples. In our setting, the goal is to separate all features within the mapped hypersphere so that no information is compressed and the feature distribution information is maximally preserved. Consequently, the choice of $\tau$ may influence the convergence behavior of training $f_{\text{cont}}$.

$$\mathcal{L}_{\text{cont}}(\mathbf{o}_{L,i}, \mathcal{S}_{\text{neg}}; f_{\text{cont}}) = \log \mathbb{E}_{\mathbf{o}_{L,j} \in \mathcal{S}_{\text{neg}}}\left[e^{\frac{1}{\tau}\cos(f_{\text{cont}}(\mathbf{o}_{L,i}), f_{\text{cont}}(\mathbf{o}_{L,j}))}\right], \quad \mathbf{o}_{L,j} \neq \mathbf{o}_{L,i}, \quad (28)$$

To assess the impact of temperature, we conduct experiments with different temperature values on the CIFAR-100 dataset using the CLIP backbone and MoE-Adapter baseline, as shown in Table 8. Since the contrastive model is lightweight and converges quickly, its output is stable across temperature settings. Moreover, because the contrastive loss is used for feature similarity ranking, the temperature scaling does not affect the relative ordering. As a result, the temperature has negligible impact on performance, and we omit this term in our implementation.

## D.4  Impact of real-synthetic feature distribution shift in CLIP model

As discussed in Section 3.3, generating inputs through model inversion solely from integer labels cannot reliably recover the rich semantic content of the feature space. This is because multiple feature representations can yield similarly low classification loss, given that the feature is the output of the penultimate layer. Consequently, the feature distribution of the synthesized data may differ from that of real data.

When synthetic features deviate from the real data distribution, this deviation reflects a shift in semantic meaning. Since CLIP models encode rich semantics in the feature space, such feature shifts correspond to substantial semantic changes in the synthetic data. Training on these misaligned samples—where labels no longer align with the underlying semantics—can degrade both the pretrained knowledge of CLIP models and the task-specific knowledge learned from previous tasks, leading to more severe forgetting and reduced zero-shot performance.

To assess the impact of performing model inversion based on classification loss, we conduct experiments on the CIFAR-100 dataset using MoE-Adapter [61] as the baseline, with all experimental settings consistent with Section 4.1. In this setup, model inversion updates the input to minimize the classification loss of the target class, whereas our method performs inversion from image features sampled from a Gaussian feature distribution and filtered by the contrastive model, as described in Section 3.3. In addition to evaluating continual learning (CL) performance, we also assess the zero-shot performance of the trained model on Tiny-ImageNet. The results, shown in Table 9, indicate that inversion based on classification loss leads to degradation in both CL and zero-shot performance, demonstrating that real-synthetic feature distribution shifts cause severe forgetting and harm the pre-trained knowledge.

Table 9: CL and zero-shot performance of MoE-Adapter combined with our method and CE loss-based model inversion. Using CE loss-based inversion degrades both continual learning and zero-shot performance.

| Model inversion method | Avg. | Last | Zero-shot |
|---|---|---|---|
| Classification loss | 86.92 | 77.84 | 54.89 |
| **Ours** | **88.35** | **81.06** | **57.17** |

# E  Feature visualization

**Distribution shift between real data and synthetic data.** Previous works [60, 39, 11] use cross-entropy loss in the model inversion objective. In the experiment on the CIFAR-100 dataset with 10 tasks, experiment settings are kept the same as experiments in Section 4.1. We visualize the features of real and synthetic data from the new classes after training each task using t-SNE, as shown in Figure 5. The synthetic data is generated by method in R-DFCIL. In the plot, synthetic features are represented by circular dots, while real data features are shown as translucent triangular dots.

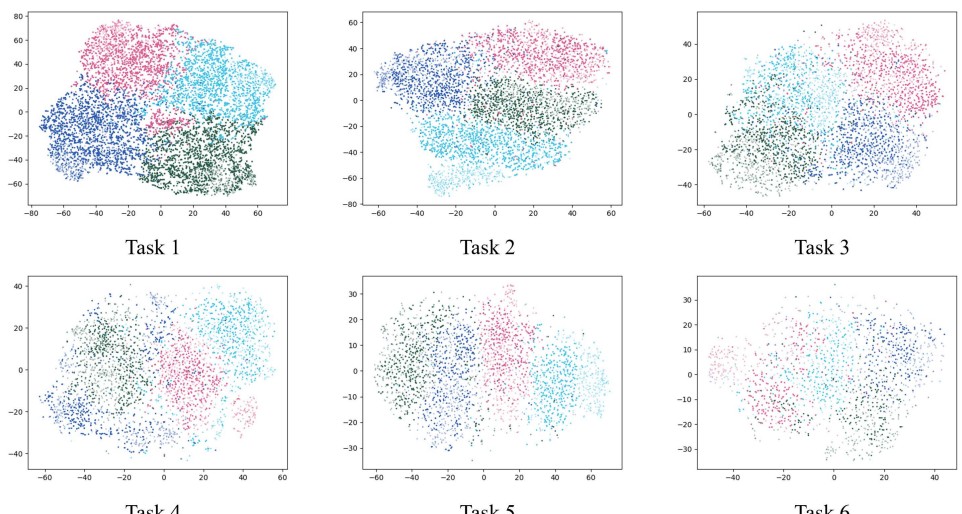

Figure 5: t-SNE visualization of features computed from synthetic data (circular dots) and real data (translucent triangular dots) from four classes in each of the first six tasks. A noticeable distribution shift can be observed between the real and synthetic features.

As shown in Figure 5, an apparent distribution shift between real and synthetic features is observed even on newly trained tasks. This shift suggests that the synthetic data may encode different information than the real data.

**Inversion from feature modeling.** To demonstrate the effectiveness of our approach for sampling features from class-wise distributions for model inversion, we additionally visualize feature t-SNE under the same experimental setting in Section 4.1, using features sampled from class-wise Gaussian distributions, as shown in Figure 6. The distributions of real and synthetic features are more consistent across tasks, indicating that our method significantly mitigates the distribution shift problem.

To further demonstrate the effectiveness of CFS based on Gaussian distributions, we generate synthetic samples using features sampled from Gaussian and Gaussian+CFS distributions, respectively, and visualize the features of four classes from the last three tasks in Figure 7. The first row of Figure 7 corresponds to sampling from class-wise Gaussian distributions for model inversion, while the second row shows results with class-wise Gaussian distributions combined with CFS. Real features are represented by translucent triangular dots.

The synthetic features in the second row more closely align with the real class-wise features compared to those in the first row, illustrating the effectiveness of our CFS method. We acknowledge that

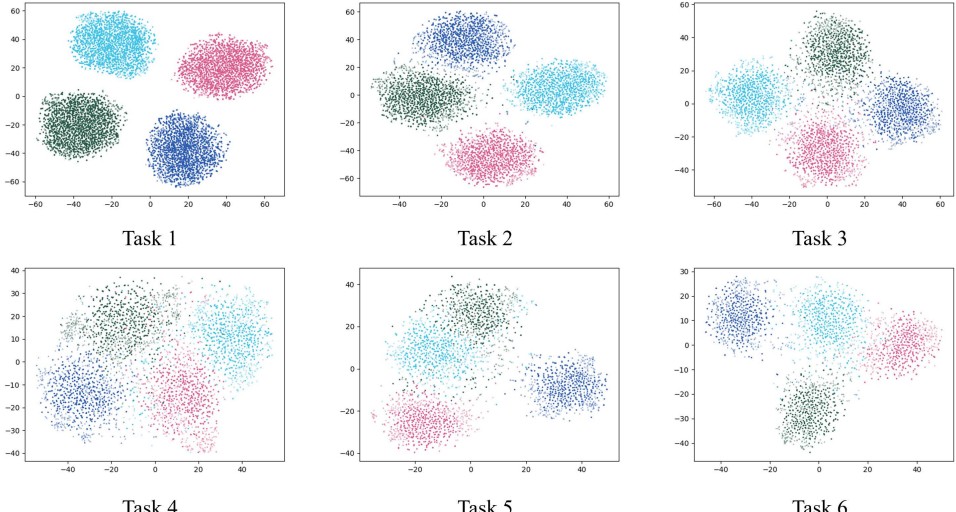

Figure 6: t-SNE visualization of features computed from synthetic data (circular dots) and real data (translucent triangular dots) from four classes in each of the first six tasks. The synthetic data is generated using features sampled from class-wise Gaussian distributions. The resulting feature distributions of synthetic and real data are more consistent.

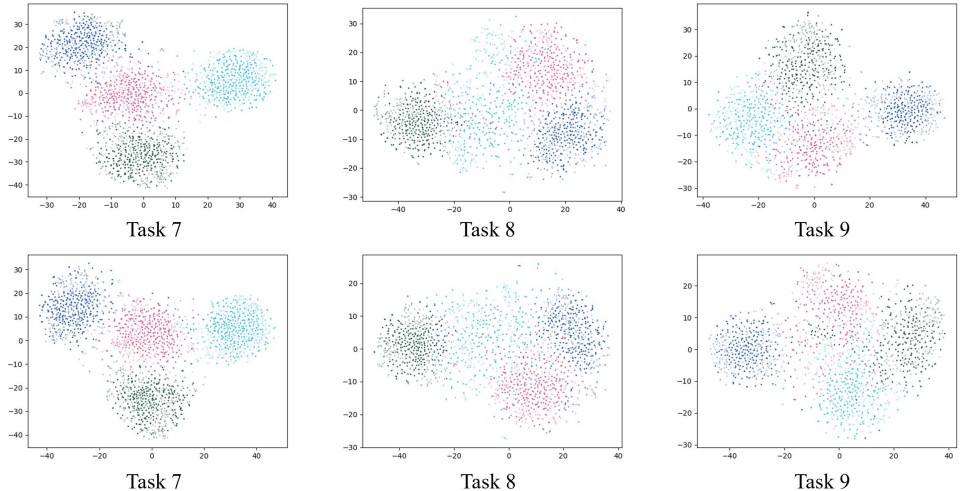

Figure 7: t-SNE visualizations of synthetic features from classes in the last three tasks. The first row shows features sampled from class-wise Gaussian distributions for model inversion, while the second row incorporates class-wise Gaussian sampling with CFS. In both cases, the synthetic and real feature distributions are more consistent, with the synthetic features in the second row more closely covering the real feature distribution.

t-SNE visualizations may not fully reflect the impact of CFS due to the significant dimensionality reduction involved. However, the results presented in Section 4.1 and Section 4.2 clearly demonstrate the effectiveness of CFS in improving CL performance.

## F    Comparison with other model inversion methods

### F.1    Noise-initialization-based methods

Noise-initialization methods start from random noise as input. DeepInversion [60] updates such inputs through full-model inversion, while Sparse Model Inversion (SMI) [14] builds on this approach by progressively pruning unimportant patches during inversion. In contrast, our method initializes

Table 10: CL performance and computational cost of different model inversion methods integrated with MoE-Adapter on the CIFAR-100 dataset. Our method achieves the best performance while incurring the lowest time cost.

| Method | Avg. | Last | Time cost | Per-image time cost |
|--------|------|------|-----------|---------------------|
| Full-model+MoE-Adapter | 88.13 | 80.53 | 17h 17m | 23.15s |
| SMI+MoE-Adapter | 88.16 | 80.46 | 7h 14m | 13.35s |
| Ours+MoE-Adapter | **88.35** | **81.06** | **6h 10m** | **7.42s** |

Table 11: CL performance of different model inversion methods integrated with MoE-Adapter on the CIFAR-100 dataset under the same iteration budget. Our method achieves the best performance.

| Method | Avg. | Last |
|--------|------|------|
| Full-model+MoE-Adapter | 88.06 | 80.09 |
| SMI+MoE-Adapter | 88.13 | 80.31 |
| Ours+MoE-Adapter | **88.35** | **81.06** |

inputs with PMI, thereby reducing the number of iterations required for full-model inversion. To evaluate the effectiveness of our method, we implement full-model inversion and SMI [14] in a CLIP-based CL setting and integrate them with MoE-Adapter [61].

Specifically, we compare our method against SMI and full-model inversion in terms of CL performance and computational cost. All experimental settings are kept consistent with those in Section 4.1, with CFS applied across all experiments. The results are presented in Table 10. To ensure fair comparison, we use the recommended settings for each baseline. For full-model inversion, we follow the settings of [19] and set the number of update steps to 3,400. For SMI, we adopt the original configuration with 4,000 update steps and a pruning ratio of 76%. Our method applies 200 steps for PMI and 600 steps for full-model inversion. All experiments are conducted on NVIDIA GeForce RTX 3090 GPUs with 24 GB of memory, using an Intel(R) Core(TM) i9-12900K CPU.

Our PMI combined with full-model inversion consistently outperforms other model inversion methods in CL performance while incurring the lowest time cost, demonstrating the effectiveness of our approach. While SMI achieves comparable time efficiency, excessive patch pruning can lead to a performance drop compared to full-model inversion. The time cost for generating a single sample further highlights the efficiency of our method.

We also provide a comparison under the same inversion iteration budget in Table 11, where all methods use 800 iterations. While the performance of baseline methods drops with the reduced budget, our method consistently outperforms them, further demonstrating its effectiveness.

### F.2 Generator-based methods

Generator-based methods train a generator via model inversion and then use it to produce samples for replay. As noted in ABD [39] and R-DFCIL [11], these approaches are generally more efficient than noise-initialization methods. To evaluate the effectiveness of our approach, we further compare our method with generator-based methods in terms of performance, time cost, and GPU efficiency.

We first compare our method with generator-based approaches, including ABD, R-DFCIL, and DCMI [33], on the CIFAR-100 dataset using a ResNet-32 backbone. The classes are evenly divided into 5 tasks, and all experimental settings follow those in Section 4.1. We report final average accuracy to evaluate performance, and GPU memory usage and FLOPs during inversion to assess efficiency. The results are presented in Table 12.

As shown in Table 12, our method achieves the best performance while requiring only slightly more time than R-DFCIL. Because the samples are independent during model inversion, GPU parallelism can be utilized more effectively. We therefore adopt a larger batch size for inversion, which results in comparable time cost to R-DFCIL and faster execution than ABD, albeit with higher GPU memory usage and FLOPs. We further note that the time cost of our method can be reduced by scaling to multiple GPUs, where samples can be generated in parallel without synchronization overhead.

Table 12: Comparison with generator-based methods in terms of performance, time cost, and GPU efficiency on the ResNet-32 backbone. Our method achieves higher performance while maintaining comparable time cost and GPU efficiency.

| Method | Last | Time | FLOPs ($10^{12}$) | GPU memory (GB) |
|---|---|---|---|---|
| ABD | 48.84±0.33 | 1h 33m | 940.35 | **3.71** |
| R-DFCIL | 49.87±0.45 | **1h 07m** | **567.32** | **3.71** |
| DCMI | 41.05±0.67 | 1h 28m | 887.59 | 6.05 |
| Ours | **52.38±0.53** | 1h 10m | 1055.70 | 7.28 |

Table 13: Comparison with generator-based methods in terms of performance, time cost, and GPU efficiency on the CLIP backbone. Our method achieves higher performance and greater GPU efficiency.

| Method | Avg. | Last | Time | FLOPs ($10^{14}$) | GPU memory (GB) |
|---|---|---|---|---|---|
| Generator-based | 87.30 | 79.97 | 7h 25m | 148.66 | 10.90 |
| Ours | **88.35** | **81.06** | **6h 10m** | **108.28** | **7.39** |

We further compare our method with generator-based approaches using the CLIP backbone on CIFAR-100. For the generator-based method, we employ a larger generator with 3.3M parameters. All inversion methods are combined with MoE-Adapter [61], and the experimental settings are consistent with Section 4.1. The results, including performance, time cost, GPU FLOPs, and GPU memory usage, are reported in Table 13.

As shown in Table 13, our method outperforms generator-based approaches and achieves superior GPU efficiency. Compared to experiments with the ResNet-32 backbone, generating only a small amount of synthetic data (5 samples per class in our setting) is sufficient to mitigate forgetting. While generator-based methods are efficient with small backbones, they struggle to capture the rich knowledge encoded in large pre-trained models. Even with a larger generator (containing more parameters than in the ResNet-32 experiments) and 6,000 training steps, generator-based inversion fails to deliver significant improvements over the baseline. These findings highlight the effectiveness of our proposed method.

While generator-based methods are more efficient at generating samples after training, noise-initialization methods offer two key advantages:

1. Since input samples are independent, they can be generated in parallel across multiple GPUs without synchronization. This allows image-noise-based methods to scale more efficiently with additional GPUs, further reducing the overall time cost.

2. When using knowledge distillation, teacher logits can be computed in a single forward pass after sample generation for image-noise-based methods. In contrast, generator-based methods generate a synthetic batch and compute teacher logits at each training step, requiring an extra forward pass in every iteration.

Our PMI method reduces the number of iterations through better initialization, and can further leverage these advantages.

## G Loss landscape visualization

To support our claim that the inversion loss landscape of a single layer is simpler than that of the full model, we implement our method on the image encoder (ViT-B/16) of the CLIP model and visualize the inversion loss landscape of each layer and the full model near the optimal point, as shown in Figure 8. The inversion loss landscapes of individual layers are significantly simpler and flatter compared to that of the full model. As a result, performing inversion through a single layer enables faster convergence and achieves lower inversion loss.

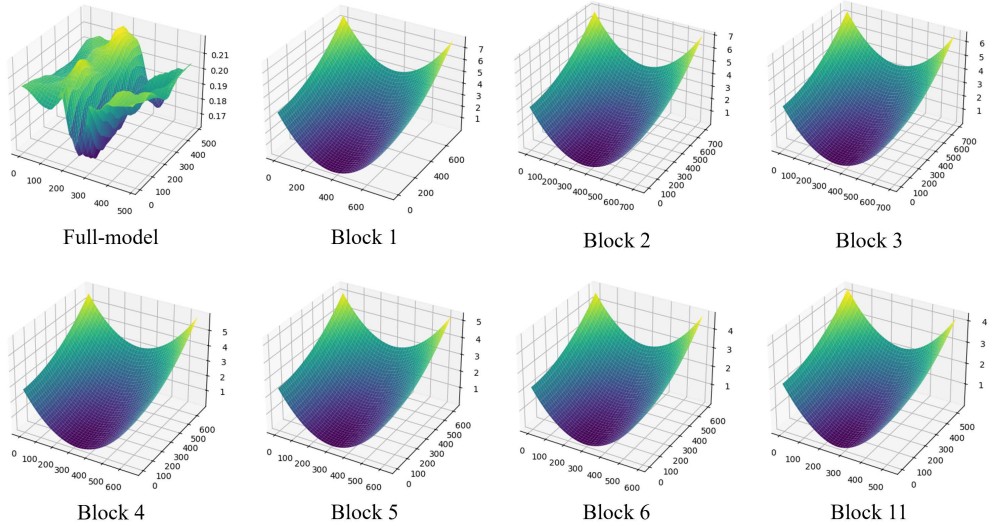

Figure 8: Inversion loss landscapes of full-model fine-tuning and individual layers. The loss landscape of a single layer is significantly simpler than that of the full model.

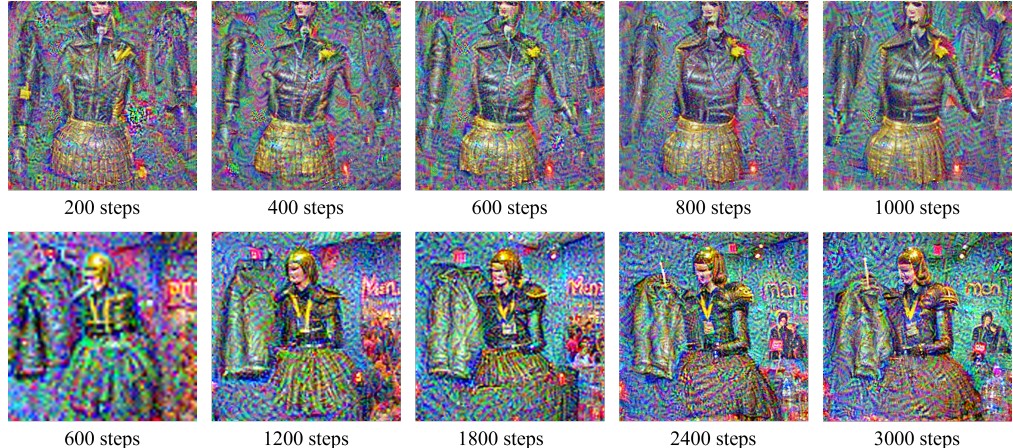

Figure 9: Images generated from the prompt *"A female mannequin dressed in a black leather jacket and gold pleated skirt."* during the model inversion process. The first row shows images generated using our PMI + full-model inversion strategy, while the second row presents images generated by the baseline method.

## H    Generating image from text using model inversion on CLIP model

To visually demonstrate the effectiveness of our PMI+full-model inversion strategy, we follow the experimental setup of [19] and present images generated during the model inversion process. Figure 9 shows images generated from the prompt: *"A female mannequin dressed in a black leather jacket and gold pleated skirt."* and Figure 10 shows images generated from the prompt: *"A big dog chasing a small kitten."* In both figures, the first row displays results produced by our PMI+full-model inversion strategy, while the second row shows results generated by the baseline method from [19]. The label *steps* in each figure indicates the number of update steps during full-model inversion. Note that images from our method are initialized using PMI.

In both examples, our method generates meaningful images with fewer update steps, highlighting the effectiveness of the PMI+full-model inversion strategy. We note, however, that generating images from text does not directly reflect CL performance. This visualization is intended to illustrate that our method significantly reduces the number of update steps required during model inversion, while still capturing key semantic features efficiently.

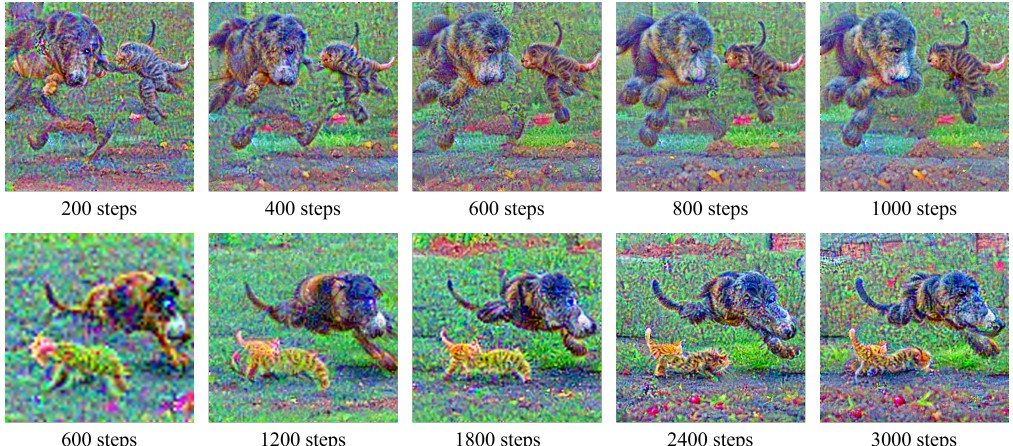

|  | 200 steps | 400 steps | 600 steps | 800 steps | 1000 steps |
|  | 600 steps | 1200 steps | 1800 steps | 2400 steps | 3000 steps |

Figure 10: Images generated from the prompt *"A big dog chasing a small kitten."* during the model inversion process. The first row shows images produced using our PMI + full-model inversion strategy, while the second row displays images generated by the baseline method.

Table 14: Base and incremental samples on different settings in ResNet-based CL experiments.

|   | 5 task. | 10 task | 20 task |
|---|---------|---------|---------|
| $a$ | 5,000 | 2,000 | 1,000 |
| $b$ | 6,000 | 3,000 | 2,000 |

# I   Experiment details

## I.1   ResNet-based CL

**Continual training.** The CIFAR-100 dataset contains 100 classes, which we evenly divide into 5, 10, and 20 disjoint tasks. Similarly, the Tiny-ImageNet dataset, consisting of 200 classes, is split into 5, 10, and 20 disjoint tasks. The class order for CL follows the setting used in R-DFCIL [11], and data augmentation includes random cropping and random horizontal flipping. The backbone model is ResNet-32 following the settings of R-DFCIL.

Each task takes totally 120 epochs for training, and learning rate is set to 0.01 with 0.1 times decreasing after 60 and 90 epochs, we use SGD optimizer for all the experiments. Loss factors for hard knowledge distillation, relational knowledge distillation and classification head finetuning loss are set to 0.15, 0.5 and 1.5 respectively.

**Model inversion.** We maintain an incremental buffer containing $a \cdot t + b$ synthetic images, where $t$ denotes the task index, $a$ is the number of new samples generated for each task, and $b$ represents the base samples. The configurations for different settings are provided in Table 14. Buffer samples are generated prior to training each new task using the model trained on the previous task. The scaling factors for the MSE loss and the distribution constraint are set to 1.0 and 0.25, respectively. For model inversion, we use the Adam optimizer, with a learning rate of 0.8 for PMI and 0.4 for full-model inversion. The number of update steps is set to 50 for PMI and 160 for full-model inversion.

**Contrastive model.** The contrastive model is a two-layer MLP, with each layer consisting of 64 neurons and Leaky ReLU as the activation function. It is trained for 200 epochs using the SGD optimizer with a learning rate of 0.01. For CFS, we set the selection ratio to 0.5 and the number of selection steps to 40.

## I.2   CLIP-based CL

**Continual training.** In CLIP-based CL, we conduct experiments on CIFAR-100, ImageNet-R, and CUB-200. Both ImageNet-R and CUB-200 contain 200 classes, and we evenly split all three datasets into 10 disjoint tasks. Following the setting of PROOF [68], we use a fixed random seed (1993) to generate the class order. The optimization parameters are summarized in Table 15. For all methods, the loss weight for the hard knowledge distillation (HKD) loss is set to 0.1. For MoE-Adapter, we additionally use a loss weight of 0.2 for the text knowledge distillation loss and 0.001 for the text

Table 15: Training hyper-parameters for CLIP-based CL, where LR denotes learning rate.

| Method | CIFAR-100 | | | ImageNet-R | | | CUB200 | | |
|---|---|---|---|---|---|---|---|---|---|
| | LR | Epoch | Optimizer | LR | Epoch | Optimizer | LR | Epoch | Optimizer |
| VPT | 0.02 | 5 | SGD | 0.01 | 5 | SGD | 0.02 | 5 | SGD |
| CODA-P | 0.1 | 5 | SGD | 0.1 | 5 | SGD | 0.1 | 5 | SGD |
| MoE-Adapter | 0.001 | 3 | AdamW | 0.001 | 3 | AdamW | 0.001 | 10 | AdamW |

encoder fine-tuning loss on CIFAR-100 and ImageNet-R. For the fine-grained dataset CUB-200, we set both the text knowledge distillation and text encoder fine-tuning loss weights to 0.1.

**Model inversion.** We maintain 5 synthetic samples for each old class. For model inversion, we use 200 update steps for PMI and 600 steps for full-model inversion. Following the setting in [19], we use the Adam optimizer and set the learning rate to 0.1 for PMI and 0.01 for full-model inversion.

In experiments on CIFAR-100 and ImageNet-R, the loss weights for the MSE loss, distribution constraint, and smoothness constraint are set to $1.0$, $2 \times 10^{-3}$, and $5 \times 10^{-3}$, respectively. For the fine-grained dataset CUB-200, the loss weights are set to $1.0$ for the MSE loss, $0.2$ for the distribution constraint, and $5 \times 10^{-3}$ for the smoothness constraint.

**Contrastive model.** The contrastive model is implemented as a two-layer MLP, where each layer contains 512 neurons and uses Leaky ReLU as the activation function. It is trained for 200 epochs using the SGD optimizer with a learning rate of 0.01. For CFS, we use a selection ratio of 0.5 and perform 5 selection steps.

### I.3 Semantic-aware feature projection

For both the CIFAR-100 and ImageNet-R datasets, we apply feature projection using the top 5 most similar classes, where similarity is measured by the cosine similarity between class text features. The parameter $\alpha$ is set to 0.1 for both datasets.

## J Broader impact

In real-world applications, data availability is a major concern when deploying machine learning techniques. Model inversion addresses this issue effectively and is widely used in data-free scenarios, including data-free knowledge transfer, data-free meta-learning, and data-free continual learning. Additionally, model inversion can be employed to analyze what a model has learned, contributing to the development of trustworthy AI systems.

Model inversion generates data by extracting knowledge from a trained model. Applying model inversion to large pre-trained models allows for better utilization of the rich knowledge encoded in these models. Our work improves the efficiency of model inversion on large pre-trained CLIP models and demonstrates the potential for continually adapting models to new classes without requiring additional data collection—by recovering data directly from the pre-trained models. Furthermore, our method addresses the issue of feature distribution shift in model inversion-based continual learning, which can help reduce forgetting and improve overall performance.

We acknowledge that model inversion may be potentially use for recovering data containing unexpected privacy appeared in training data, which could be a systematic issue of whole pipeline of data collection, training, model deployment, and inversion. However, our method is designed to improve the efficiency of model inversion and to better model class-wise feature distributions, rather than to recover private information. Moreover, model inversion relies on the knowledge encoded in the model; without privacy-related information being encoded, our method cannot recover any private data.

## K Computation resources and asset URLs

All experiments are conducted on a system with four NVIDIA GeForce RTX 3090 GPUs (24 GB each) and an Intel(R) Core(TM) i9-12900K CPU with 64 GB of RAM.

Our experiments include the CIFAR-100, Tiny-ImageNet, ImageNet-R, and CUB-200 datasets. The URLs for these datasets are:

- `https://www.cs.toronto.edu/~kriz/cifar.html`
- `https://github.com/hendrycks/imagenet-r`
- `https://www.vision.caltech.edu/datasets/cub_200_2011/`

Our ResNet-based CL experiments are implemented based on the open-source code of R-DFCIL [11] and DCMI [33]. The URLs for these implementations are:

- `https://github.com/jianzhangcs/R-DFCIL`
- `https://github.com/zihuanqiu/DCMI_CVPR24`

Our CLIP-based CL experiments are implemented based on the open-source code of PROOF [68], CODA-Prompt [40], and MoE-Adapter [61]. The URLs for these implementations are:

- `https://github.com/LAMDA-CL/PROOF`
- `https://github.com/GT-RIPL/CODA-Prompt`
- `https://github.com/JiazuoYu/MoE-Adapters4CL`

The implementation of CLIP model inversion is based on [19]. The code URL is:

- `https://github.com/hamidkazemi22/CLIPInversion`

We confirm that all assets used in our work, including datasets, code, and pre-trained models, are used in accordance with their respective licenses.

