# OpenReview forum: "Model Inversion with Layer-Specific Modeling and Alignment for Data-Free Continual Learning"
_NeurIPS.cc/2025/Conference — NeurIPS 2025 poster_

### Official Review · Reviewer_wMRU · 2025-06-29

**Clarity:** 3
**Significance:** 3
**Originality:** 3
**Rating:** 4
**Confidence:** 4

**Summary:**

This paper proposes PMI, a method designed to improve model inversion by providing better initialization, thereby accelerating the inversion process. In addition, the authors introduce CFS, a sampling strategy that selects informative and diverse class-wise features from Gaussian distributions. Experiments conducted on both ResNet and CLIP models demonstrate the effectiveness of the proposed methods and validate the claimed improvements.

**Questions:**

see weakness.

**Ethical Concerns:**

["NO or VERY MINOR ethics concerns only"]

**Final Justification:**

Although the innovation is not fundamentally groundbreaking, the paper presents a clearly written and practically meaningful approach. By leveraging hierarchical optimization, it effectively achieves better initialization and reduces training overhead. Combined with the novel use of model inversion for unseen class data generation and consistent improvements over baselines, the method offers a valuable and applicable contribution to continual learning.

**Limitations:**

yes

**Quality:**

3

**Strengths And Weaknesses:**

Pros:

(1) The paper is clearly written, and the proposed method effectively addresses the stated motivation.

(2) Leveraging model inversion to generate data for unseen classes is a novel and potentially promising direction in the field of continual learning.

(3) The proposed method demonstrates satisfactory improvements over existing baselines.

Cons:

(1) The comparison of inversion time cost is not sufficiently comprehensive. Table 8 only reports results for the Full-model and SMI. Including additional baselines would better justify the claimed efficiency of PMI. Moreover, the number of inversion iterations should be controlled across methods to fairly assess performance under limited iterations.

(2) While PMI accelerates inversion via better initialization, methods like ABD and DCMI use random-initialized generators to improve speed and reduce memory usage. A comparison with these methods in terms of inversion time and GPU efficiency is missing. Discussing the differences between image-noise-based and generator-based approaches would better support the motivation.

---

> ### Author Rebuttal · Authors · 2025-07-31
>
> Thank you for your constructive and insightful review, these suggestions have helped us identify and strengthen important aspects of our experimental analysis.
>
>
> **Weaknesses**
>
> **W1.1: Including more runtime comparison.**
>
> **A1.1:** We have included a comparison of the time cost of our method with generator-based model inversion baselines (e.g., ABD, R-DFCIL, DCMI) using the ResNet-32 backbone on the CIFAR-100 5-task setting, as shown in Table below. While our method requires more time, it achieves superior performance. Additionally, we compare our method with a generator-based approach using the CLIP backbone on CIFAR-100 with the MoE-Adapter baseline. As shown in Table below, our method achieves higher accuracy with lower runtime.
>
> |Method| Last | Time | FLOPs (10^12) | GPU memory (GB) |
> |---------|:----:|:----:|:-:|:-:|
> | ABD     |48.84±0.33|1h 33m|940.35|3.71|
> | R-DFCIL |49.87±0.45|1h 07m|567.32|3.71|
> | DCMI |41.05±0.67|1h 28m|887.59|6.05|
> | Our Method |52.38±0.53|1h 10m|1055.70|7.28|
>
> |Method|Avg.| Last | Time |FLOPs (10^14)|GPU memory (GB)|
> |---------|:-:|:----:|:----:|:-:|:-:|
> |Generator+MoE-Adapter|87.30|79.97|7h 25m|148.66|10.90|
> |Ours+MoE-Adapter|88.35|81.06|6h 10m|108.28|7.39|
>
> **W1.2: Control iteration steps for performance comparison.**
>
> **A1.2** For the full-model inversion and SMI baselines, we originally used their default iteration settings to ensure optimal performance. As shown in Table below, we re-evaluate all methods using a fixed budget of 800 iteration steps. Under this constraint, the performance of both baselines drops noticeably, further demonstrating the effectiveness of our method within the same iteration budget.
>
> | Method | Avg. | Last |
> |-|:----:|:----:|
> | Full-model+MoE-Adapter |88.06|80.09|
> | SMI+MoE-Adapter |88.13|80.31|
> | Ours+MoE-Adapter |88.35|81.06|
>
> **W2.1: GPU efficiency comparison.**
>
> **A2.1** We have included GPU FLOPs and memory usage for data generation in each task in Tables above. Our method requires more FLOPs and GPU memory when using the ResNet-32 backbone. However, with the CLIP backbone, our method is more efficient using fewer FLOPs and less memory, since generating only a small amount of synthetic data (5 samples per class in our experiments) is sufficient to mitigate forgetting. While generator-based methods are efficient with small backbones, they struggle to effectively capture the knowledge encoded in large pre-trained models. Even with a larger generator (containing more parameters than those used in the ResNet-32 experiments) and 6,000 training steps, generator-based inversion fails to significantly improve performance over the baseline. These results demonstrate the effectiveness of our proposed method.
>
> **W2.2: Discussing differences between image-noise-based and generator-based methods.**
>
> **A2.2** While generator-based methods are more efficient at generating samples after training, image-noise-based methods offer two key advantages:
>
> 1. Since input samples are independent, they can be generated in parallel across multiple GPUs without synchronization. This allows image-noise-based methods to scale more efficiently with additional GPUs, further reducing the overall time cost.
> 2. When using knowledge distillation, teacher logits can be computed in a single forward pass after sample generation for image-noise-based methods. In contrast, generator-based methods generate a synthetic batch and compute teacher logits at each training step, requiring an extra forward pass in every iteration.
>
> Our PMI method reduces the number of iterations through better initialization, and can further leverage these advantages.
>
> Thank you again for your thoughtful comments. We will incorporate all the analyses and discussions into the revised version of our paper.

---

> > ### Comment · Reviewer_wMRU · 2025-08-05
> >
> > Thanks for authors' rebuttal. My concerns are addressed.

---

> > > ### Author Response · Authors · 2025-08-05
> > >
> > > Thank you for your response! We’re glad that our rebuttal addressed your concerns, and we appreciate your thoughtful review for our work.

---

### Official Review · Reviewer_VxGQ · 2025-06-30

**Clarity:** 3
**Significance:** 3
**Originality:** 3
**Rating:** 4
**Confidence:** 4

**Summary:**

The authors proposed two techniques to improve the efficiency and effectiveness of the model inversion-based method in data-free continual learning. Specifically, they leverage a Per-layer Model Inversion (PMI) approach inspired by the faster convergence of single-layer optimization, which reduces the number of iterations required for convergence. Moreover, they model class-wise feature distribution using a Gaussian distribution and sampling features from them for inversion. By combining PMI and feature modeling, the proposed method demonstrates efficiency and effectiveness in multiple incremental settings.

**Questions:**

Apart from the concerns in the “Weakness” part, In Eq. (6), there is no temperature \tau inside the \exp function as the contrastive loss. I was wondering whether it may affect the convergence.

**Ethical Concerns:**

["NO or VERY MINOR ethics concerns only"]

**Final Justification:**

Thanks for the detailed rebuttal. All my concerns and questions have been well addressed. Especially, the supplemented experiments have further shown the superiority of the proposed method. I will maintain my score for acceptance recommendation.

**Limitations:**

The authors adequately addressed the limitations and potential negative societal impact of their work.

**Paper Formatting Concerns:**

There are no major formatting issues in this paper.

**Quality:**

3

**Strengths And Weaknesses:**

Strengths:
1. Comprehensive experiments have been conducted to verify the effectiveness and efficiency of the proposed method (different incremental settings including ResNet- and CLIP-based CIL, ablation studies, etc.).
2. Although the work follows the branch of model inversion-based data-free continual learning, which is not as vibrant as Non-Exemplar CIL probably due to the extra time for synthesizing images and rehearsal, the proposed techniques have certain novelty.
3. The paper is generally well written with a clear logic flow. The methodology and the findings are presented in a well-structured manner.

Weaknesses:
1. Since synthesizing images via model inversion may incur much time, a runtime analysis (comparison with other methods such as ABD, and R-DFCIL) should better be included. In Sec. F of the supplementary, there is a runtime analysis that does not cover the aforementioned methods.
2. Although not strongly related to model inversion, recent works in Non-Exemplar CIL should be discussed due to its similar data-free formulation (e.g. [A1][A2]).
[A1] Kai Zhu, Wei Zhai, Yang Cao, Jiebo Luo, Zheng-Jun Zha. Self-Sustaining Representation Expansion for Non-Exemplar Class-Incremental Learning. CVPR 2022
[A2] Wuxuan Shi, Mang Ye. Prototype Reminiscence and Augmented Asymmetric Knowledge Aggregation for Non-Exemplar Class-Incremental Learning. ICCV 2023

Others that may not affect the rating:
1. “ci” and “cj” are not clear indices for classes. Better use one character.
2. In Fig. 1, the order of the steps is arranged from right to left, which affects the readability.
3. In Table 4, a value of “-0.15” is in bold, which is not supposed to be that way according to my understanding.
4. Result tables use different types of visualization to highlight the best performance. Better use the same.

---

> ### Author Rebuttal · Authors · 2025-07-31
>
> Thank you for your constructive and insightful review, these suggestions have helped us identify and strengthen important aspects of our experimental analysis.
>
> **Weaknesses**
>
> **W1: Including more runtime comparison.**
>
> **A1:** We have compared the training time of our method with generator-based model inversion baselines (e.g., ABD, R-DFCIL, DCMI) using the ResNet-32 backbone on the CIFAR-100 5-task setting, as shown in Table below. While our method requires more running time, it achieves superior performance. Additionally, we compare our approach to a generator-based method using the CLIP backbone on CIFAR-100 with the MoE-Adapter baseline. As shown in Table below, our method achieves higher accuracy with lower runtime. Moreover, unlike generator-based methods, our approach treats each sample independently, allowing for efficient parallel generation across multiple GPUs, which can further reduce the overall time cost.
>
> | Method | Last | Time |
> |---------|:----:|:----:|
> | ABD     |48.84±0.33|1h 33m|
> | R-DFCIL |49.87±0.45|1h 07m|
> | DCMI |41.05±0.67|1h 28m|
> | Our Method |52.38±0.53|1h 10m|
>
> | Method |Avg.| Last | Time |
> |---------|:-:|:----:|:----:|
> |Generator+MoE-Adapter|87.30|79.97|7h 25m|
> |Ours+MoE-Adapter|88.35|81.06|6h 10m|
>
> **W2: Missing non-exemplar CIL baselines.**
>
> **A2:** Thank you for your comments. We have included these baselines in our experimental setup on the CIFAR-100 dataset using the ResNet-32 backbone for additional comparison, as shown in Table below. We will also include these baselines in the revised version of our paper.
>
> | Method | 5 task | 10 task | 20 task |
> |-------|:------:|:-------:|:-------:|
> | SSRE  |30.39±0.04|17.77±0.14|10.97±0.48|
> | PRAKA |37.74±0.48|26.72±0.38|16.32±0.66|
> |Our method|52.38±0.53|43.90±0.35|32.60±0.29|
>
> **W3: Minor clarity issues.**
>
> **A3:** Thank you for your suggestions! We will make the following revisions in the next version of our paper:
> 1. simplify the notation of class indices,
> 2. change the layout of Figure 1 to a left-to-right orientation,
> 3. correct the wrong bold formatting in Table 4,
> 4. ensure a consistent style across all performance tables.
>
> **Questions**
>
> **Q1: Ablation study on temperature.**
>
> **A1:** We conducted experiments with different temperature values on the CIFAR-100 dataset using the CLIP backbone and MoE-Adapter baseline, as shown in Table below. Since the contrastive model is lightweight and converges quickly, its output is stable across temperature settings. Moreover, because the contrastive loss is used for feature similarity ranking, the temperature scaling does not affect the relative ordering. As a result, the temperature has negligible impact on performance, and we omit this term in our implementation.
>
> | Temperature | Avg. | Last |
> |:-----------:|:----:|:----:|
> | 0.5 |88.26|80.96|
> | 0.8 |88.27|81.16|
> | 1.0 |88.35|81.06|
> | 1.5 |88.24|80.94|

---

> > ### Comment · Reviewer_VxGQ · 2025-08-07
> >
> > Thanks for the detailed rebuttal. All my concerns and questions have been well addressed. Especially, the supplemented experiments have further shown the superiority of the proposed method. I will maintain my score for acceptance recommendation.

---

> > > ### Author Response · Authors · 2025-08-07
> > >
> > > Thank you for your thoughtful response and encouraging feedback! We’re glad that our rebuttal and additional experiments address your concerns. We sincerely appreciate your support and constructive suggestions throughout the review process.

---

### Official Review · Reviewer_ZqGB · 2025-07-02

**Clarity:** 2
**Significance:** 2
**Originality:** 2
**Rating:** 4
**Confidence:** 3

**Summary:**

This paper proposes a method to improve data-free continual learning based on model inversion. It tackles two key challenges: the high computational cost of inversion and the feature drift between synthetic data coming from model inversion and real data. To improve efficiency, the authors introduce Per-layer Model Inversion (PMI), which generates a strong initialization for the inversion process by optimizing layer-by-layer, thus reducing the total number of required iterations. To mitigate feature drift, they model the class-wise feature distributions using Gaussian and contrastive models to guide the generation of more realistic synthetic samples. The paper uses CLIP as the backbone, using the original pre-trained model as a generator to create synthetic training data for classes that are new to the continual learning task.

**Questions:**

See weaknesses.

**Ethical Concerns:**

["NO or VERY MINOR ethics concerns only"]

**Final Justification:**

Thanks for the thorough rebuttal which adresses some of my concerns. I've increased my score.

**Limitations:**

yes

**Paper Formatting Concerns:**

No concerns

**Quality:**

2

**Strengths And Weaknesses:**

Weaknesses:
- The paper argues that the problems of data-free continual learning (i.e. synthetic / real drift + expensive model inversion) are even more severe in models like CLIP (line 14). This is not clear to me why that would be the case. Why would CL be worse in CLIP models compared to any other model ? There's no empirical nor theoretical argument given by the authors and so the motivation to work with CLIP is weak.
Also CLIP is already a powerful general-purpose model that has seen data similar to imagenet or CIFAR during its pretraining. It already has some knowledge about the "unseen" categories (as we can see from the strong zero-shot capabilities) and so we cannot really say that these categories are "unseen".

- Experimental results are not very convincing. The results shown in Table 1 for example are only marginally above the results from R-DFCIL, which is a method which is 3 years old.

- To generate new synthetic data for replay, the method uses the original, frozen pre-trained model. This is less an "inversion" of the learning model and more akin to using a powerful, external generative model. This setup blurs the lines of the contribution and deviates from the core data-free continual learning problem where knowledge should be extracted from the model being trained. If one can use the original frozen CLIP, why not leveraging any other foundation models (i.e. text to image generative models) to generate data for continual learning ?

- The method's reliance on storing "input statistics from real data" for each layer fundamentally challenges its "data-free" claim. The assumption that one cannot store raw data but can compute and store detailed, layer-wise statistics derived from it seems unrealistic. Can the authors comment on this ? Is there a way to make the method work without this assumption ?

---

> ### Author Rebuttal · Authors · 2025-07-31
>
> We sincerely thank you for your detailed and thoughtful review, these comments have been particularly helpful for enhancing and clarifying our work.
>
> **Weaknesses**
>
> **W1.1: Motivation to work with CLIP model.**
>
> **A1:** As we claimed in **line 8-11**, generating samples with model inversion solely from integer labels can lead to a feature distribution gap between real and synthetic data. This issue is particularly pronounced in the CLIP feature space, which encodes rich semantic information. When synthetic features drift from the real data distribution, it implies a shift in semantic meaning. Training on such misaligned samples where the label corresponds to incorrect semantics can degrade both the pre-trained knowledge of CLIP and the task-specific knowledge learned from previous tasks, resulting in more severe forgetting. We conduct an experiment on CIFAR-100 using classification loss for model inversion with the MoE-adapter as the baseline, and report the results in Table below. The performance is worse than the non-replay baseline (**79.40 in Table 2**), highlighting a more severe synthetic/real drift issue in the CLIP setting. Moreover, model inversion on CLIP requires significantly more computation and more iterations to converge compared to smaller backbones (e.g., ResNet). Thus, both the feature drift and computational cost are more substantial challenges when applying model inversion to CLIP models.
>
> | Method | Avg. | Last |
> |------------|:----:|:----:|
> | CE inversion |86.92 |77.84 |
> | Our method |88.35 |81.06 |
>
> **W1.2: “Unseen” class definition is unclear.**
>
> **A1.2:** In section 3.4 of our work, “unseen” classes refer to those that the model has not been adapted to during training. While CLIP models may have been pre-trained on semantically similar classes, our method demonstrates the feasibility of generating data for specific target classes and improving performance on them through adaptation using the generated samples. We will revise our work to provide a clearer definition.
>
> **W2: Improvement over baseline.**
>
> **A2:** R-DFCIL is a strong baseline in model inversion for continual learning settings. As shown in Table 5 of Appendix D.1, the more recent method DCMI (2024) does not outperform R-DFCIL under our experimental setup. Our method provides less performance improvement on 20-task settings which is a challenging setting for all data-free continual learning methods, and our method consistently outperforms R-DFCIL one all the settings, demonstrating the effectiveness of our method.
>
> **W3: Model for generating data.**
>
> **A3:** The CLIP-based experiments in Table 2 follow the standard data-free setting, where replay data is generated from the continually trained model. In the experiments shown in Figure 2, we assume that images from the last task are unavailable (unseen), and generate them using the original CLIP model, since the continually trained model may have lost some pre-trained knowledge. Replay data for earlier tasks is still generated by the continually trained model. Therefore, our experimental setup adheres to the standard formulation of the data-free continual learning problem.
>
> **W4: Storing input statistics.**
>
> **A4:** Thank you for your comments! First, unlike previous inversion methods for data-free continual learning that rely solely on layer-specific statistics from BatchNorm layers, our method maintains a small set of additional statistics—specifically, the mean and standard deviation for each data set. These statistics are highly compressed and do not reveal any original data, making the approach both practical and compliant with the data-free setting. Storing such minimal statistics does not violate the data-free assumption, and using BatchNorm statistics is a common and accepted practice in prior model inversion works. Importantly, storing input statistics is not the main contribution of our method. Since CLIP is pre-trained on a highly diverse dataset, it can support model inversion even without relying on stored statistics. To verify this, we conducted experiments on CIFAR-100 using the MoE-Adapter baseline without input statistics. As shown in Table below, performance dropped noticeably, especially on the fine-grained CUB-200 dataset. These results indicate the importance of input-statistics alignment for maintaining high-quality inversion.
>
> |  Method  | Dataset | Avg. | Last |
> |---------------------------------|:-------:|:----:|:----:|
> | Our method w/o input statistics |CIFAR-100|88.17 |80.71 |
> | Our method                      |CIFAR-100|88.35 |81.06 |
> | Our method w/o input statistics |CUB-200  |77.75 |66.33 |
> | Our method                      |CUB-200  |78.98 |67.26 |
>
> We appreciate the reviewer’s feedback and will include all related discussions and clarifications in the revised version.

---

> > ### Author Response · Authors · 2025-08-08
> > **A kind reminder**
> >
> > Dear Reviewer ZqGB,
> >
> > Thank you again for your thoughtful review. We sincerely appreciate the time and effort you’ve dedicated to evaluating our submission. We would like to kindly follow up, in case there are any remaining questions or clarifications needed from our side. We hope our rebuttal has addressed your concerns, and we are happy to provide any further details and clarification.
> >
> > Best regards,
> > Authors of submission 1225.

---

### Official Review · Reviewer_8o8v · 2025-07-03

**Clarity:** 3
**Significance:** 3
**Originality:** 3
**Rating:** 4
**Confidence:** 3

**Summary:**

This paper addresses the inefficiency issue of model inversion for data-free continual learning. It proposes a PMI technique to efficiently generate synthetic data by performing model inversion layer by layer, thus reducing computational cost and improving convergence. To mitigate feature drift between synthetic and real data, they model class-wise feature distributions and use these for sampling features in inversion. Experimental results on some datasets validate the effectiveness of the method, which improves the classification accuracy and training efficiency.

**Questions:**

See wesknesses.

**Ethical Concerns:**

["NO or VERY MINOR ethics concerns only"]

**Final Justification:**

Thank you for the authors' responses. My concerns have been largely addressed, and I maintain my score as Borderline Accept.

**Limitations:**

Yes

**Quality:**

3

**Strengths And Weaknesses:**

Strengths:

1.	The proposed PMI presents an elegant idea of decomposing the inversion process, leading to faster convergence and lower computational cost, which is also justified theoretically and empirically.

2.	The combination of Gaussian modeling and contrastive feature selection improves the quality of synthetic data.

Weaknesses:

1.	Though PMI reduces the number of update steps, the extra step of per-layer inversion adds overhead. It will be better to compare the training time of the proposed method with baselines.

2.	The effectiveness of CFS seems minor, as shown in Table 1.

3.	The limitations of the proposed method are not discussed comprehensively.

---

> ### Author Rebuttal · Authors · 2025-07-31
>
> Thank you for your constructive feedback and valuable suggestions, these points have helped us better clarify and improve the presentation and evaluation of our method.
>
> **Weaknesses**
>
> **W1: Computation overhead of PMI.**
>
> **A1:** We have compared the time cost of our method with full-model inversion and Sparse Model Inversion (SMI) in Appendix F. Our method uses 200 steps for PMI and 600 for full-model inversion. In contrast, the full-model inversion baseline requires 3,400 steps and SMI requires 4,000 steps. Our method achieves better performance while requiring less total time than these baselines. These results indicate that PMI introduces lower overhead and substantially reduces the total number of iterations. Furthermore, PMI allows for larger batch sizes due to lower GPU memory usage per layer, enabling more efficient utilization of GPU parallelism. As a result, the extra initialization cost can be amortized, reducing the per-sample computation time.
>
> **W2: Performance improvement of CFS.**
>
> **A2:** Our motivation for using CFS is to better preserve feature distribution information compared to Gaussian modeling. In our experiments, CFS consistently improves performance, including when applied to the CLIP backbone, as shown in **Tables 3 and 4**. These results support our insight that preserving richer distributional information through contrastive modeling can lead to further performance gains. Additionally, CFS employs a lightweight MLP to better capture the true feature distribution, incurring only minimal computational overhead.
>
> **W3: Limitations not comprehensively discussed.**
>
> **A3:** Due to space limitations, we primarily discussed the limitation that more sophisticated methods could be employed for generating data of unseen classes. We will expand on this and provide a more detailed discussion of limitations in the revised version.

---

> > ### Comment · Reviewer_8o8v · 2025-08-05
> >
> > Thank you for the authors' responses. My concerns have been largely addressed, and I maintain my score as Borderline Accept.

---

> > > ### Author Response · Authors · 2025-08-05
> > >
> > > Thank you for your follow-up and for acknowledging that our rebuttal has addressed your concerns. We appreciate your constructive review, which has helped us improve our work.

---

### Decision · Program_Chairs · 2025-09-17

**Decision:**

Accept (poster)

**Comment:**

The paper proposes Per-layer Model Inversion (PMI) and Contrastive Feature Selection (CFS) for data-free continual learning, offering an elegant decomposition of inversion that accelerates convergence and modestly improves performance. Strengths lie in the conceptual clarity and novelty of PMI, consistent improvements across settings, and solid rebuttal clarifications. However, the empirical gains are small, CFS adds marginal benefit, and reliance on CLIP and stored statistics weakens the “data-free” framing. While not groundbreaking and with limited experimental persuasiveness and huge memory and compute overhead (in terms of FLOPs compared to prior works), the work is technically sound and contributes an interesting direction.